# Understanding Negative Samples in Instance Discriminative Self-supervised Representation Learning

**Kento Nozawa**
The University of Tokyo & RIKEN AIP
nzw@g.ecc.u-tokyo.ac.jp

**Issei Sato**
The University of Tokyo
sato@g.ecc.u-tokyo.ac.jp

## Abstract

Instance discriminative self-supervised representation learning has been attracted attention thanks to its unsupervised nature and informative feature representation for downstream tasks. In practice, it commonly uses a larger number of negative samples than the number of supervised classes. However, there is an inconsistency in the existing analysis; theoretically, a large number of negative samples degrade classification performance on a downstream supervised task, while empirically, they improve the performance. We provide a novel framework to analyze this empirical result regarding negative samples using the coupon collector's problem. Our bound can implicitly incorporate the supervised loss of the downstream task in the self-supervised loss by increasing the number of negative samples. We confirm that our proposed analysis holds on real-world benchmark datasets.

## 1 Introduction

Self-supervised representation learning is a popular class of unsupervised representation learning algorithms in the domains of vision [Bachman et al., 2019, Chen et al., 2020a, He et al., 2020, Caron et al., 2020, Grill et al., 2020, Chen and He, 2021] and language [Mikolov et al., 2013, Devlin et al., 2019, Brown et al., 2020]. Generally, it trains a feature extractor by solving a pretext task constructed on a large unlabeled dataset. The learned feature extractor yields generic feature representations for other machine learning tasks such as classification. Recent self-supervised representation learning algorithms help a linear classifier to attain classification accuracy comparable to a supervised method from scratch, especially in a few amount of labeled data regime [Newell and Deng, 2020, Hénaff et al., 2020, Chen et al., 2020b]. For example, `SwAV` [Caron et al., 2020] with ResNet-50 has a top-1 validation accuracy of $75.3\%$ on the ImageNet-1K classification [Deng et al., 2009] compared with $76.5\%$ by using the fully supervised method.

InfoNCE [van den Oord et al., 2018, Eq. 4] or its modification is a de facto standard loss function used in many state-of-the-art self-supervised methods [Logeswaran and Lee, 2018, Bachman et al., 2019, He et al., 2020, Chen et al., 2020a, Hénaff et al., 2020, Caron et al., 2020]. Intuitively, the minimization of InfoNCE can be viewed as the minimization of cross-entropy loss on $K + 1$ instance-wise classification, where $K$ is the number of negative samples. Despite the empirical success of self-supervised learning, we still do not understand why the self-supervised learning algorithms with InfoNCE perform well for downstream tasks.

Arora et al. [2019] propose the first theoretical framework for contrastive unsupervised representation learning (CURL). However, there exists a gap between the theoretical analysis and empirical observation as in Figure 1. Precisely, we expect that a large number of negative samples degrade a supervised loss on a downstream task from the analysis by Arora et al. [2019]. In practice, however,

a large number of negative samples are commonly used in self-supervised representation learning algorithms [He et al., 2020, Chen et al., 2020a].

**Contributions.** We show difficulty to explain why large negative samples empirically improve supervised accuracy on the downstream task from the CURL framework when we use learned representations as feature vectors for the supervised classification in Section 3. To fill the gap, we propose a novel lower bound to theoretically explain this empirical observation regarding negative samples using the coupon collector's problem in Section 4.

## 2 InfoNCE-based Self-supervised Representations Learning

We focus on Chen et al. [2020a]'s self-supervised representation learning formulation, namely, `SimCLR`.[1] Let $\mathcal{X}$ be an input space, e.g., $\mathcal{X} \subset \mathbb{R}^{\text{channel} \times \text{width} \times \text{height}}$ for color images. We can only access a unlabeled training dataset $\{\mathbf{x}_i\}_{i=1}^N$, where input $\mathbf{x} \in \mathcal{X}$. `SimCLR` learns feature extractor $\mathbf{f} : \mathcal{X} \to \mathbb{R}^h$ modeled by neural networks on the dataset, where $h$ is the dimensionality of feature representation. Let $\mathbf{z} = \mathbf{f}(\mathbf{a}(\mathbf{x}))$ that is a feature representation of $\mathbf{x}$ after applying data augmentation $\mathbf{a} : \mathcal{X} \to \mathcal{X}$. Data augmentation is a pre-defined stochastic function such as a composition of the horizontal flipping and cropping. Note that the output of $\mathbf{f}$ is normalized by its L2 norm. Let $\mathbf{z}^+ = \mathbf{f}(\mathbf{a}^+(\mathbf{x}))$ that is a positive feature representation created from $\mathbf{x}$ with different data augmentation $\mathbf{a}^+(\cdot)$.

`SimCLR` minimizes the following InfoNCE-based loss with $K$ negative samples for each pair of $(\mathbf{z}, \mathbf{z}^+)$:

$$\ell_{\text{Info}}(\mathbf{z}, \mathbf{Z}) \coloneqq -\ln \frac{\exp\left(\mathbf{z} \cdot \mathbf{z}^+/t\right)}{\sum_{\mathbf{z}_k \in \mathbf{Z}} \exp\left(\mathbf{z} \cdot \mathbf{z}_k/t\right)}, \tag{1}$$

where $\mathbf{Z} = \{\mathbf{z}^+, \mathbf{z}_1^-, \ldots, \mathbf{z}_K^-\}$ that is a set of

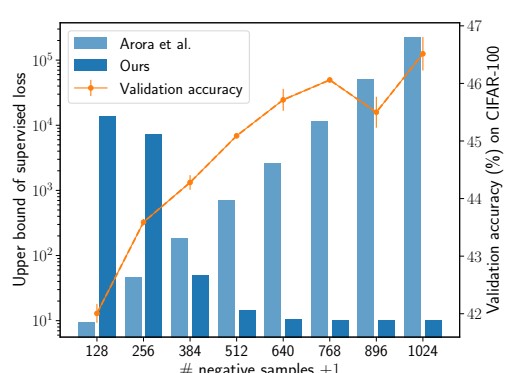

Figure 1: Upper bounds of supervised loss and validation accuracy on CIFAR-100. By increasing the number of negative samples, validation accuracy tends to improve. **Left bars**: The existing contrastive unsupervised representation learning bound (8) also increases when the number of negative samples increases because the bound of supervised loss explodes due to a collision term that is not related to classification loss (see Eq. (8) for the definition). **Right bars with ★**: On the other hand, the proposed counterpart (10) does not explode. Section 5 describes the details of this experiment.

positive representation $\mathbf{z}^+$ and $K$ negative representations $\{\mathbf{z}_1^- \ldots, \mathbf{z}_K^-\}$ created from other samples, $(\cdot \cdot)$ is an inner product of two representations, and $t \in \mathbb{R}_+$ is a temperature parameter.[2] Intuitively, this loss function approximates an instance-wise classification loss by using $K$ random negative samples [Wu et al., 2018]. From the definition of Eq. (1), we expect that $\mathbf{f}$ learns an invariant encoder with respect to data augmentation $\mathbf{a}$. After minimizing Eq. (1), $\mathbf{f}$ works as a feature extractor for downstream tasks such as classification.

## 3 Extension of Contrastive Unsupervised Representation Learning Framework

We focus on the direct relationship between the self-supervised loss and a supervised loss to understand the role of the number of negative samples $K$. For a similar problem, the CURL framework [Arora et al., 2019] shows that the averaged supervised loss is bounded by the contrastive unsupervised loss. Thus, we extend the analysis of CURL to the self-supervised representation learning in Section 2 and point out that we have difficulty explaining the empirical observation as

---

[1] In fact, our theoretical analysis is valid with asymmetric feature extractors such as `MoCo` [He et al., 2020], where the positive and negative features do not come from feature extractor $\mathbf{f}$.

[2] We perform the analysis with $t = 1$ for simplicity, but our analysis holds with any temperature.

in Figure 1 from the existing analysis. Table 2 in Appendix A summarizes notations used in this paper for convenience.

## 3.1 CURL Formulation for `SimCLR`

By following the CURL analysis [Arora et al., 2019], we introduce the learning process in two steps: *self-supervised representation learning step* and *supervised learning step* (see Section 3.1.1 and Section 3.1.2, respectively).

### 3.1.1 Self-supervised representation learning step

The purpose of the self-supervised learning step is to learn a feature extractor $\mathbf{f}$ on an unlabeled dataset. During this step, we can only access input samples from $\mathcal{X}$ without any relationship between samples, unlike metric learning [Kulis, 2012] or similar unlabeled learning [Bao et al., 2018].

We formulate the data generation process for Eq. (1). The key idea of the CURL analysis is the existence of latent classes $\mathcal{C}$ that are associated with supervised classes.[3] Let $\rho$ be a probability distribution over $\mathcal{C}$, and let $\mathbf{x}$ be an input sample drawn from a data distribution $\mathcal{D}_c$ conditioned on a latent class $c \in \mathcal{C}$. We draw two data augmentations $(\mathbf{a}, \mathbf{a}^+)$ from the distribution of data augmentation $\mathcal{A}$ and apply them independently to the sample. As a result, we have two augmented samples $(\mathbf{a}(\mathbf{x}), \mathbf{a}^+(\mathbf{x}))$ and call $\mathbf{a}^+(\mathbf{x})$ as a positive sample of $\mathbf{a}(\mathbf{x})$. Similarly, we draw $K$ negative samples from $\mathcal{D}_{c_k^-}$ for each $c_k^- \in \{c_1^-, \ldots, c_K^-\} \sim \rho^K$ and $K$ data augmentations from $\mathcal{A}$, and then we apply them to negative samples. Definition 1 summarizes the data generation process. Note that we suppose data augmentation $\mathbf{a} \sim \mathcal{A}$ does not change the latent class of $\mathbf{x} \in \mathcal{X}$.

**Definition 1** (Data Generation Process in Self-supervised Representation Learning Step).
*1. Draw latent classes: $c, \{c_k^-\}_{k=1}^K \sim \rho^{K+1}$;    2. Draw input sample: $\mathbf{x} \sim \mathcal{D}_c$;*
*3. Draw data augmentations: $(\mathbf{a}, \mathbf{a}^+) \sim \mathcal{A}^2$;    4. Apply data augmentations: $\mathbf{a}(\mathbf{x}), \mathbf{a}^+(\mathbf{x})$;*
*5. Draw negative samples: $\{\mathbf{x}_k^-\}_{k=1}^K \sim \mathcal{D}_{c_k^-}^K$;    6. Draw data augmentations: $\{\mathbf{a}_k^-\}_{k=1}^K \sim \mathcal{A}^K$;*
*7. Apply data augmentations: $\{\mathbf{a}_k^-(\mathbf{x}_k^-)\}_{k=1}^K$.*

Note that the original data generation process of CURL samples a positive sample of $\mathbf{x}$ from $\mathcal{D}_c$ independently and does not use any data augmentations.

From the data generation process and InfoNCE loss (1), we give the following formal definition of self-supervised loss.

**Definition 2** (Expected Self-supervised Loss).

$$L_{\mathrm{Info}}(\mathbf{f}) := \mathop{\mathbb{E}}_{c, \{c_k^-\}_{k=1}^K \sim \rho^{K+1}} \mathop{\mathbb{E}}_{\substack{\mathbf{x} \sim \mathcal{D}_c \\ (\mathbf{a}, \mathbf{a}^+) \sim \mathcal{A}^2}} \mathop{\mathbb{E}}_{\substack{\{\mathbf{x}_k^- \sim \mathcal{D}_{c_k^-}\}_{k=1}^K \\ \{\mathbf{a}_k^-\}_{k=1}^K \sim \mathcal{A}^K}} \ell_{\mathrm{Info}}(\mathbf{z}, \mathbf{Z}), \tag{2}$$

*where recall that $\mathbf{z} = \mathbf{f}(\mathbf{a}(\mathbf{x}))$ and $\mathbf{Z} = \{\mathbf{f}(\mathbf{a}^+(\mathbf{x})), \mathbf{f}(\mathbf{a}_1^-(\mathbf{x}_1^-)), \ldots, \mathbf{f}(\mathbf{a}_K^-(\mathbf{x}_K^-))\}$.*

Since we cannot directly minimize Eq. (2), we minimize the empirical counterpart, $\widehat{L}_{\mathrm{Info}}(\mathbf{f})$, by sampling from the training dataset. After minimizing $\widehat{L}_{\mathrm{Info}}(\mathbf{f})$, learned $\widehat{\mathbf{f}}$ works as a feature extractor in the supervised learning step.

### 3.1.2 Supervised learning step

At the supervised learning step, we can observe label $y \in \mathcal{Y} = \{1, \ldots, Y\}$ for each $\mathbf{x}$ that is used in the self-supervised learning step. Let $\mathcal{S}$ be a supervised data distribution over $\mathcal{X} \times \mathcal{Y}$, and $\mathbf{g} \circ \widehat{\mathbf{f}} : \mathcal{X} \to \mathbb{R}^Y$ be a classifier, where $\mathbf{g} : \mathbb{R}^h \to \mathbb{R}^Y$ and $\widehat{\mathbf{f}}$ is the frozen feature extractor. Given the

---

[3]Technically, we do not need $c$ to draw $\mathbf{x}$ for self-supervised representation learning. However, we explicitly include it in the data generation process to understand the relationship with a supervised task in Section 3.2. In the main analysis, we assume that the supervised classes in the downstream task are subset of $\mathcal{C}$; however, different relationship between supervised and latent classes is discussed in Appendix B.

supervised data distribution and classifier, our goal is to minimize the following supervised loss:

$$L_{\text{sup}}(\mathbf{g} \circ \widehat{\mathbf{f}}) := \mathop{\mathbb{E}}_{\substack{\mathbf{x},y \sim \mathcal{S} \\ \mathbf{a} \sim \mathcal{A}}} - \ln \frac{\exp\left(\mathbf{g}_y(\widehat{\mathbf{f}}(\mathbf{a}(\mathbf{x})))\right)}{\sum_{j \in \mathcal{Y}} \exp\left(\mathbf{g}_j(\widehat{\mathbf{f}}(\mathbf{a}(\mathbf{x})))\right)}. \tag{3}$$

By following the CURL analysis, we introduce a mean classifier as a simple instance of $\mathbf{g}$ because its loss is an upper bound of Eq. (3).[4] The mean classifier is the linear classifier whose weight of label $y$ is computed by averaging representations: $\boldsymbol{\mu}_y = \mathbb{E}_{\mathbf{x} \sim \mathcal{D}_y} \mathbb{E}_{\mathbf{a} \sim \mathcal{A}} \mathbf{f}(\mathbf{a}(\mathbf{x}))$, where $\mathcal{D}_y$ is a data distribution conditioned on the supervise label $y$. We introduce the definition of the mean classifier's supervised loss as follows:

**Definition 3** (Mean Classifier's Supervised Loss).

$$L_{\text{sup}}^{\mu}(\widehat{\mathbf{f}}) := \mathop{\mathbb{E}}_{\substack{\mathbf{x},y \sim \mathcal{S} \\ \mathbf{a} \sim \mathcal{A}}} - \ln \frac{\exp\left(\widehat{\mathbf{f}}(\mathbf{a}(\mathbf{x})) \cdot \boldsymbol{\mu}_y\right)}{\sum_{j \in \mathcal{Y}} \exp\left(\widehat{\mathbf{f}}(\mathbf{a}(\mathbf{x})) \cdot \boldsymbol{\mu}_j\right)}. \tag{4}$$

We also introduce a sub-class loss function.[5] Let $\mathcal{Y}_{\text{sub}}$ be a subset of $\mathcal{Y}$ and $\mathcal{S}_{\text{sub}}$ be a data distribution over $\mathcal{X} \times \mathcal{Y}_{\text{sub}}$, then we define the sub-class losses of classifier $\mathbf{g}$ and mean classifier:

**Definition 4** (Supervised Sub-class Losses of Classifier $\mathbf{g}$ and Mean Classifier with $\widehat{\mathbf{f}}$).

$$L_{\text{sub}}(\mathbf{g} \circ \widehat{\mathbf{f}}, \mathcal{Y}_{\text{sub}}) := \mathop{\mathbb{E}}_{\substack{\mathbf{x},y \sim \mathcal{S}_{\text{sub}} \\ \mathbf{a} \sim \mathcal{A}}} - \ln \frac{\exp\left(\mathbf{g}_y(\widehat{\mathbf{f}}(\mathbf{a}(\mathbf{x})))\right)}{\sum_{j \in \mathcal{Y}_{\text{sub}}} \left(\mathbf{g}_j(\widehat{\mathbf{f}}(\mathbf{a}(\mathbf{x})))\right)}, \tag{5}$$

$$L_{\text{sub}}^{\mu}(\widehat{\mathbf{f}}, \mathcal{Y}_{\text{sub}}) := \mathop{\mathbb{E}}_{\substack{\mathbf{x},y \sim \mathcal{S}_{\text{sub}} \\ \mathbf{a} \sim \mathcal{A}}} - \ln \frac{\exp\left(\widehat{\mathbf{f}}(\mathbf{a}(\mathbf{x})) \cdot \boldsymbol{\mu}_y\right)}{\sum_{j \in \mathcal{Y}_{\text{sub}}} \exp\left(\widehat{\mathbf{f}}(\mathbf{a}(\mathbf{x})) \cdot \boldsymbol{\mu}_j\right)}. \tag{6}$$

The purpose of unsupervised representation learning [Bengio et al., 2013] is to learn generic feature representation rather than improve the accuracy of the classifier on the same dataset. However, we believe that such a feature extractor tends to transfer well to another task. Indeed, Kornblith et al. [2019] empirically show a strong correlation between ImageNet's accuracy and transfer accuracy.

## 3.2 Theoretical Analysis based on CURL

We show that InfoNCE loss (2) is an upper bound of the expected sub-class loss of the mean classifier (6).

**Step 1. Introduce a lower bound** We denote $\boldsymbol{\mu}(\mathbf{x}) = \mathbb{E}_{\mathbf{a} \sim \mathcal{A}} \mathbf{f}(\mathbf{a}(\mathbf{x}))$ and derive a lower bound of unsupervised loss $L_{\text{Info}}(\mathbf{f})$.

$$L_{\text{Info}}(\mathbf{f}) \geq \mathop{\mathbb{E}}_{c,\{c_k^-\}_{k=1}^K \sim \rho^{K+1}} \mathop{\mathbb{E}}_{\substack{\mathbf{x} \sim \mathcal{D}_c \\ \mathbf{a} \sim \mathcal{A}}} \mathop{\mathbb{E}}_{\{\mathbf{x}_k^- \sim \mathcal{D}_{c_k^-}\}_{k=1}^K} \ell_{\text{Info}}\left(\mathbf{f}(\mathbf{a}(\mathbf{x})), \{\boldsymbol{\mu}(\mathbf{x}), \boldsymbol{\mu}(\mathbf{x}_1^-), \ldots, \boldsymbol{\mu}(\mathbf{x}_K^-)\}\right)$$

$$\geq \mathop{\mathbb{E}}_{c,\{c_k^-\}_{k=1}^K \sim \rho^{K+1}} \mathop{\mathbb{E}}_{\substack{\mathbf{x} \sim \mathcal{D}_c \\ \mathbf{a} \sim \mathcal{A}}} \ell_{\text{Info}}\left(\mathbf{f}(\mathbf{a}(\mathbf{x})), \left\{\boldsymbol{\mu}(\mathbf{x}), \boldsymbol{\mu}_{c_1^-}, \ldots, \boldsymbol{\mu}_{c_K^-}\right\}\right)$$

$$\geq \mathop{\mathbb{E}}_{c,\{c_k^-\}_{k=1}^K \sim \rho^{K+1}} \mathop{\mathbb{E}}_{\substack{\mathbf{x} \sim \mathcal{D}_c \\ \mathbf{a} \sim \mathcal{A}}} \ell_{\text{Info}}\left(\mathbf{f}(\mathbf{a}(\mathbf{x})), \left\{\boldsymbol{\mu}_c, \boldsymbol{\mu}_{c_1^-}, \ldots, \boldsymbol{\mu}_{c_K^-}\right\}\right) + d(\mathbf{f}), \tag{7}$$

$$\text{where } d(\mathbf{f}) = \frac{1}{t} \mathop{\mathbb{E}}_{c \sim \rho} \mathop{\mathbb{E}}_{\mathbf{x} \sim \mathcal{D}_c} \left\{ \mathop{\mathbb{E}}_{\mathbf{a} \sim \mathcal{A}} [\mathbf{f}(\mathbf{a}(\mathbf{x}))] \cdot \left[ \mathop{\mathbb{E}}_{\substack{\mathbf{x}^+ \sim \mathcal{D}_c \\ \mathbf{a}_1^+ \sim \mathcal{A}}} \left[\mathbf{f}(\mathbf{a}_1^+(\mathbf{x}^+))\right] - \mathop{\mathbb{E}}_{\mathbf{a}_2^+ \sim \mathcal{A}} \left[\mathbf{f}(\mathbf{a}_2^+(\mathbf{x}))\right] \right] \right\}.$$

---

[4]Concrete inequality is found in Appendix C.
[5]Arora et al. [2019] refer to this loss function as *averaged supervised loss*.

The first and second inequalities are done by using Jensen's inequality for convex function. The proof of the third inequality is shown in Appendix D.1. It is worth noting that positive and negative features can be extracted from another feature encoder or memory back He et al. [2020].

**Remark 5** (Effect of gap term $d(\mathbf{f})$). *We confirm that $d(\mathbf{f})$ is an almost constant among different $K$ in practice (see Table 1). Therefore we focus on the first term in Eq. (7) in the following analysis.*

**Step 2. Decomposition into the expected sub-class loss** We convert the first term of Eq. (7) into the expected sub-class loss explicitly. By following Arora et al. [2019, Theorem B.1], we introduce collision probability: $\tau_K = \mathbb{P}(\mathrm{Col}(c, \{c_k^-\}_{k=1}^K) \neq 0)$, where $\mathrm{Col}(c, \{c_k^-\}_{k=1}^K) = \sum_{k=1}^K \mathbb{I}[c = c_k^-]$ and $\mathbb{I}[\cdot]$ is the indicator function. We omit the arguments of Col for simplicity. Let $\mathcal{C}_{\mathrm{sub}}(\{c, c_1^-, \ldots, c_K^-\})$ be a function to remove duplicated latent classes given latent classes. We omit the arguments of $\mathcal{C}_{\mathrm{sub}}$ as well. The result is our extension of Arora et al. [2019, Lemma 4.3] for self-supervised representation learning as the following proposition:

**Proposition 6** (CURL Lower Bound of Self-supervised Loss). *For all feature extractor $\mathbf{f}$,*

$$L_{\mathrm{Info}}(\mathbf{f}) \geq (1 - \tau_K) \underset{c, \{c_k^-\}_{k=1}^K \sim \rho^{K+1}}{\mathbb{E}} [\underbrace{L_{\mathrm{sub}}^{\mu}(\mathbf{f}, \mathcal{C}_{\mathrm{sub}})}_{\text{sub-class loss}} \mid \mathrm{Col} = 0]$$

$$+ \tau_K \underset{c, \{c_k^-\}_{k=1}^K \sim \rho^{K+1}}{\mathbb{E}} [\underbrace{\ln(\mathrm{Col} + 1)}_{\text{collision}} \mid \mathrm{Col} \neq 0] + d(\mathbf{f}). \tag{8}$$

The proof is found in Appendix D.2.

Arora et al. [2019] also show Rademacher complexity-based generalization error bound for the mean classifier. Since we focus on the behavior of the self-supervised loss rather than a generalization error bound when $K$ increases, we do not perform further analysis as was done in Arora et al. [2019]. Nevertheless, we believe that constructing a generalization error bound by following either Arora et al. [2019, Theorem 4.1] or Nozawa et al. [2020, Theorem 7] is worth interesting future work.

## 3.3 Limitations of Eq. (8)

The bound (8) tells us that $K$ gives a trade-off between the sub-class loss and collision term via $\tau_K$. Recall that our final goal is to minimize the supervised loss (4) rather than expected sub-class loss (6). To incorporate the supervised loss (4) into the lower bound (8), we need $K$ large enough to satisfy $\mathcal{C}_{\mathrm{sub}} \supseteq \mathcal{Y}$. However, such a large $K$ makes the lower bound meaningless.

We can easily observe that the lower bound (8) converges to the collision term by increasing $K$ since the collision probability $\tau_K$ converges to 1. As a result, the sub-class loss rarely contributes to the lower bound. Let us consider the bound on CIFAR-10, where the number of supervised classes is 10, and latent classes are the same as the supervised classes. When $\tau_{K=32} \approx 0.967$, i.e., the only 3.3% training samples contribute to the expected sub-class loss, the others fall into the collision term. Indeed, Arora et al. [2019] show that small latent classes or large negative samples degrade classification performance of the expected sub-class classification task. However, even much larger negative samples, $K + 1 = 512$, yield the best performance of a linear classifier with self-supervised representation on CIFAR-10 [Chen et al., 2020a, B.9].

# 4 Proposed Lower Bound for Instance-wise Self-supervised Representation Learning

To fill the gap between the theoretical bound (8) and empirical observation from recent work, we propose another lower bound for self-supervised representation learning. The key idea of our bound is to replace $\tau$ with a different probability since $\tau$ can quickly increase depending on $K$. The idea is motivated by focusing on the supervised loss rather than the expected sub-class loss.

## 4.1 Proposed Lower Bound

Let $\upsilon_K$ be a probability that sampled $K$ latent classes contain all latent classes: $\{c_k\}_{k=1}^K \supseteq \mathcal{C}$. This probability appears in the coupon collector's problem of probability theory (e.g., Durrett [2019,

Example 2.2.7]). If the latent class probability is uniform: $\forall c, \rho(c) = 1/|\mathcal{C}|$, then we can calculate the probability explicitly as follows.

**Definition 7** (Probability to Draw All Latent Classes). *Assume that $\rho$ is a uniform distribution over latent classes $\mathcal{C}$. The probability that $K$ latent classes drawn from $\rho$ contain all latent classes is defined as*

$$v_K := \sum_{n=1}^{K} \sum_{m=0}^{|\mathcal{C}|-1} \binom{|\mathcal{C}|-1}{m} (-1)^m \left(1 - \frac{m+1}{|\mathcal{C}|}\right)^{n-1}, \tag{9}$$

*where the first summation is a probability that $n$ drawn latent samples contain all latent classes [Nakata and Kubo, 2006, Eq. 2].[6]*

By replacing $\tau$ with $v$, we obtain our lower bound of InfoNCE loss:

**Theorem 8** (Proposed Lower Bound of Self-supervised Loss). *For all feature extractor $\mathbf{f}$,*

$$L_{\text{Info}}(\mathbf{f}) \geq \frac{1}{2} \Big\{ v_{K+1} \underset{c,\{c_k^-\}_{k=1}^K \sim \rho^{K+1}}{\mathbb{E}} [\underbrace{L_{\text{sub}}^\mu(\mathbf{f}, \mathcal{C})}_{\text{sup. loss}} \mid \mathcal{C}_{\text{sub}} = \mathcal{C}]$$

$$+ (1 - v_{K+1}) \underset{c,\{c_k^-\}_{k=1}^K \sim \rho^{K+1}}{\mathbb{E}} [\underbrace{L_{\text{sub}}^\mu(\mathbf{f}, \mathcal{C}_{\text{sub}})}_{\text{sub-class loss}} \mid \mathcal{C}_{\text{sub}} \neq \mathcal{C}]$$

$$+ \underset{c,\{c_k^-\}_{k=1}^K \sim \rho^{K+1}}{\mathbb{E}} \underbrace{\ln(\text{Col} + 1)}_{\text{collision}} \Big\} + d(\mathbf{f}). \tag{10}$$

The proof is found in Appendix D.3.

Theorem 8 tells us that probability $v_{K+1}$ converges to 1 by increasing the number of negative samples $K$; as a result, the self-supervised loss is more likely to contain the supervised loss and the collision term. The sub-class loss contributes to the self-supervised loss when $K$ is small as in Eq. (8). Let us consider the example value of $v$ with the same setting discussed in Section 3.3. When $v_{K=32} \approx 0.719$, i.e., $71.9\%$ training samples contribute the supervised loss.

## 4.2 Increasing $K$ does not Spread Normalized Features within Same Latent Class

We argue that the feature representations do not have a large within-class variance on the feature space by increasing $K$ in practice. To do so, we show the upper bound of the collision term in the lower bounds to understand the effect of large negative samples.

**Corollary 9** (Upper Bound of Collision Term). *Given a latent class $c$, $K$ negative classes $\{c_k^-\}_{k=1}^K$, and feature extractor $\mathbf{f}$,*

$$\ln\left(\text{Col}\left(c, \{c_k^-\}_{k=1}^K\right) + 1\right) \leq \alpha + \beta \underset{\substack{\mathbf{x} \sim \mathcal{D}_c \\ (\mathbf{a}, \mathbf{a}^+) \sim \mathcal{A}^2}}{\mathbb{E}} \underset{\substack{\mathbf{x}' \sim \mathcal{D}_c \\ \mathbf{a}' \sim \mathcal{A}}}{\mathbb{E}} \left| \mathbf{f}(\mathbf{a}(\mathbf{x})) \cdot \left[\mathbf{f}(\mathbf{a}'(\mathbf{x}')) - \mathbf{f}(\mathbf{a}^+(\mathbf{x}))\right] \right|, \tag{11}$$

*where $\alpha$ and $\beta$ are non-negative constants depending on the number of duplicated latent classes.*

The proof is found in Appendix D.4. A similar bound is shown by Arora et al. [2019, Lemma 4.4].

Intuitively, we expect that two feature representations in the same latent class tend to be dissimilar by increasing $K$. However, Eq. (11) converges even if $K$ is small in practice. Let us consider the condition when this upper bound achieves the maximum. Since $\mathbf{f}(\mathbf{a}(\mathbf{x}))$ and $\mathbf{f}(\mathbf{a}^+(\mathbf{x}))$ are computed from the same input sample with different data augmentations, their inner product tends to be 1; thus, $\mathbf{f}(\mathbf{a}'(\mathbf{x}'))$ is located in the opposite direction of $\mathbf{f}(\mathbf{a}(\mathbf{x}))$. But, it is not possible to learn such representations because of the definition of InfoNCE with normalized representation and the dimensionality of feature space: Equilateral dimension. We confirm that Eq. (11) without $\alpha$ and $\beta$ does not increase by increasing $K$ (see Table 1 and Appendix F for more analysis).

---

[6]We show expected $K+1$ to draw all supervised labels for ImageNet-1K and all used datasets in Appendix E.

### 4.3 Small $K$ can Give Consistent Loss Function for Supervised Task

As we shown in Theorem 8, $L_{\text{Info}}$ with large $K$ can be viewed as an upper bound of $L_{\text{sup}}^{\mu}$. However, $L_{\text{Info}}$ with smaller $K$ can still yield good feature representations for downstream tasks on ImageNet-1K as reported by Chen et al. [2020a]. Arora et al. [2019] also reported similar results on CIFAR-100 with contrastive losses.[7] To shed light on this smaller $K$ regime, we focus on the class distributions of both datasets, ImageNet-1K and CIFAR-100, which are almost uniform.

**Proposition 10** (Optimality of $L_{\text{sub}}$). *Suppose $\mathcal{C} \supseteq \mathcal{Y}$ and $\rho$ is uniform: $\forall c \in \mathcal{C}, \rho(c) = 1/|\mathcal{C}|$. Suppose a constant function $\mathbf{q} : \mathbf{x} \in \mathcal{X} \overset{\mathbf{q}}{\mapsto} \left[q_1, \ldots, q_{|\mathcal{C}|}\right]^{\top}$. Optimal $\mathbf{q}^*$ is a constant function that outputs a vector with the same value if and only if it minimizes $\mathbb{E}_{c, \{c_k^-\}_{k=1}^{K} \sim \rho^{K+1}} L_{\text{sub}}(\mathbf{q}^*, \mathcal{C}_{\text{sub}})$. The optimal $\mathbf{q}^*$ is also the minimizer of $L_{\text{sup}}$.*

The proof is found in Appendix D.5 inspired by Titsias [2016, Proposition 2].

Proposition 10 *does not* argue that self-supervised representation learning algorithms fail to learn feature representations on a dataset with a non-uniform class distribution. This is because we perform a supervised algorithm on a downstream dataset after representation learning in general.

### 4.4 Relation to Clustering-based Self-supervised Representation Learning

During the minimization of $L_{\text{Info}}$, we cannot minimize $L_{\text{sup}}^{\mu}$ in Eq. (10) directly since we cannot access supervised and latent classes. Interestingly, we find a similar formulation in clustering-based self-supervised representation learning algorithms.

**Remark 11.** *Clustering-based self-supervised representation learning algorithms, such as* `DeepCluster` *[Caron et al., 2018],* `SeLa` *[Asano et al., 2020],* `SwAV` *[Caron et al., 2020], and* `PCL` *[Li et al., 2021a], use prototype representations instead of $\mathbf{z}^+, \{\mathbf{z}_k^-\}_{k=1}^{K}$ by applying unsupervised clustering on feature representations. This procedure is justified as the approximation of latent class's mean representation $\boldsymbol{\mu}_c$ with a prototype representation to minimize the supervised loss in Eq. (10) rather than Eq. (2), as a result, the mini-batch size does not depend on the number of negative samples.*

This replacement supports the empirical observation in Caron et al. [2020, Section 4.3], where the authors reported `SwAV` maintained top-1 accuracy on ImageNet-1K with a small mini-batch size, 200, compared to a large mini-batch, 4 096. On the other hand, `SimCLR` [Chen et al., 2020a] did not.

## 5 Experiments

We numerically confirm our theoretical findings by using `SimCLR` [Chen et al., 2020a] on the image classification tasks. Appendix F contains NLP experiments and some analyses that have been omitted due to the lack of space. We used datasets with a relatively small number of classes to compare bounds. Our experimental codes are available online.[8]

**Datasets and Data Augmentations** We used the CIFAR-10 and CIFAR-100 [Krizhevsky, 2009] image classification datasets with the original 50 000 training samples for both self-supervised and supervised training and the original 10 000 validation samples for the evaluation of supervised learning. We used the same data augmentations in the CIFAR-10 experiment by Chen et al. [2020a]: random resize cropping with the original image size, horizontal flipping with probability 0.5, color jitter with a strength parameter of 0.5 with probability 0.8, and grey scaling with probability 0.2.

**Self-supervised Learning** We mainly followed the experimental setting provided by Chen et al. [2020a] and its implementation.[9] We used ResNet-18 [He et al., 2016] as a feature encoder without the last fully connected layer. We replaced the first convolution layer with the convolutional layer

---

[7]Arora et al. [2019, Table D.1] mentioned that this phenomenon is not covered by the CURL framework.

[8]`https://github.com/nzw0301/Understanding-Negative-Samples`. We used Hydra [Yadan, 2019], GNU Parallel [Tange, 2020], Scikit-learn [Pedregosa et al., 2011], Pandas [Reback et al., 2020], Matplotlib [Hunter, 2007], and seaborn [Waskom, 2021] in our experiments.

[9]`https://github.com/google-research/simclr`

with 64 output channels, the stride size of 1, the kernel size of 3, and the padding size of 3. We removed the first max-pooling from the encoder, and we added a non-linear projection head to the end of the encoder. The projection head consisted of the fully connected layer with 512 units, batch-normalization [Ioffe and Szegedy, 2015], ReLU activation function, and another fully connected layer with 128 units and without bias.

We trained the encoder by using PyTorch [Paszke et al., 2019]'s distributed data-parallel training [Li et al., 2020] on four GPUs, which are NVIDIA Tesla P100 on an internal cluster. For distributed training, we replaced all batch-normalization with synchronized batch-normalization. We used stochastic gradient descent with momentum factor of 0.9 on 500 epochs. We used LARC [You et al., 2017][10], and its global learning rate was updated by using linear warmup at each step during the first 10 epochs, then updated by using cosine annealing without restart [Loshchilov and Hutter, 2017] at each step until the end. We initialized the learning rate with $(K+1)/256$. We applied weight decay of $10^{-4}$ to all weights except for parameters of all synchronized batch-normalization and bias terms. The temperature parameter was set to $t = 0.5$.

**Linear Evaluation**    We report the validation accuracy of two linear classifiers: mean classifier and linear classifier. We constructed a mean classifier by averaging the feature representations of $\mathbf{z}$ per supervised class. We applied one data augmentation to each training sample, where the data augmentation was the same as in that the self-supervised learning step. For linear classifier $\mathbf{g}$, we optimized $\mathbf{g}$ by using stochastic gradient descent with Nesterov's momentum [Sutskever et al., 2013] whose factor is 0.9 with 100 epochs. Similar to self-supervised training, we trained the classifier by using distributed data-parallel training on the four GPUs with 512 mini-batches on each GPU. The initial learning rate was 0.3, which was updated by using cosine annealing without restart [Loshchilov and Hutter, 2017] until the end.

**Bound Evaluation**    We compared the extension bound of CURL (8) and our bound (10) by varying the number of negative samples $K$. We selected $K + 1 \in \{32, 64, 128, 256, 512\}$ for CIFAR-10 and $K + 1 \in \{128, 256, 384, 512, 640, 786, 896, 1024\}$ for CIFAR-100. After self-supervised learning, we approximated $\boldsymbol{\mu}_c$ by averaging 10 sampled data augmentations per sample on the training dataset and evaluated Equations (8) and (10) with the same negative sample size $K$ as in self-supervised training.[11] We reported the averaged values over validation samples with 10 epochs: $(\lfloor 10\,000/(K+1) \rfloor \times (K+1) \times \text{epoch})$ pairs of $(\mathbf{z}, \mathbf{Z})$. Note that we used a theoretical value of $\upsilon$ defined by Eq. (9) to avoid dividing by zero if a realized $\upsilon$ value is 0. We also reported the upper bound of collision (11) without constants $\alpha, \beta$ referred to as "Collision Bound" on the training data. See Appendix F for details.

### 5.1  Experimental Results

Table 1 shows the bound values on CIFAR-10 and CIFAR-100. We only showed a part of values among different numbers of negative samples due to the page limitation.[12] We reported mean and linear classifiers' validation accuracy as "$\mu$ acc" and "Linear acc", respectively. As a reference, we reported practical linear evaluation's validation accuracy as "Linear acc w/o", where we discarded the non-linear projection head from the feature extractor [Chen et al., 2020a]. Since the CURL bound (8) does not contain $^\dagger L_{\text{sup}}^\mu$ explicitly, we subtracted $^\dagger L_{\text{sup}}^\mu$ from $L_{\text{sub}}^\mu$ in Eq. (8) and reported $^\dagger L_{\text{sup}}^\mu$ and subtracted $L_{\text{sub}}^\mu$ as $^\dagger L_{\text{sub}}^\mu$ for the comparison. We confirmed that CURL bounds converged to $^\dagger$Collision with relatively small $K$. On the other hand, proposed bound values had still a large proportion of supervised loss $L_{\text{sup}}^\mu$ with larger $K$. Figure 1 shows *upper* bounds of supervised loss $L_{\text{sup}}$ and the linear accuracy by rearranging Equations (8) and (10). The reported values were averaged over three training runs of both self-supervised and supervised steps with different random seeds. The error bars represented the standard deviation.

---

[10]https://github.com/NVIDIA/apex

[11]Precisely, $\boldsymbol{\mu}_c = \frac{1}{N_c} \sum_{i=1}^{N} \mathbb{I}[y_i = c] \frac{1}{10} \sum_{j=1}^{10} \mathbf{f}(\mathbf{a}_j(\mathbf{x}_i))$.

[12]Tables 4 and 5 in Appendix F provides the comprehensive results.

Table 1: The bound values on CIFAR-10/100 experiments with different $K+1$. CURL bound and its quantities are shown with †. The proposed ones are shown without †. Since the proposed collision values are half of †Collision, they are omitted. The reported values contain their coefficient except for Collision bound.

| $K+1$ | | CIFAR-10 | | | | CIFAR-100 | | | |
|---|---|---|---|---|---|---|---|---|---|
| | | 32 | 128 | 256 | 512 | 128 | 256 | 512 | 1024 |
| $\tau$ | | 0.96 | 1.00 | 1.00 | 1.00 | 0.72 | 0.92 | 0.99 | 1.00 |
| $\upsilon$ | | 0.69 | 1.00 | 1.00 | 1.00 | 0.00 | 0.00 | 0.62 | 1.00 |
| $\mu$ acc | | 72.75 | 77.22 | 78.60 | 80.12 | 32.67 | 34.25 | 35.90 | 37.44 |
| Linear acc | | 77.13 | 81.33 | 82.85 | 84.13 | 41.95 | 43.53 | 45.16 | 46.57 |
| Linear acc w/o | | 82.02 | 85.43 | 86.68 | 87.66 | 57.92 | 58.91 | 59.30 | 59.46 |
| $L_{\mathrm{Info}}$ | Eq. (2) | 2.02 | 3.29 | 3.96 | 4.64 | 3.32 | 3.98 | 4.66 | 5.34 |
| $d(\mathbf{f})$ | Eq. (7) | $-1.16$ | $-1.18$ | $-1.18$ | $-1.19$ | $-0.99$ | $-0.98$ | $-0.97$ | $-0.95$ |
| $^{\dagger}L_{\mathrm{Info}}$ bound | Eq. (8) | 0.23 | 1.41 | 2.08 | 2.75 | 0.72 | 0.46 | 0.78 | 1.42 |
| $^{\dagger}$Collision | | 1.32 | 2.58 | 3.26 | 3.94 | 0.69 | 1.15 | 1.73 | 2.37 |
| $^{\dagger}L^{\mu}_{\mathrm{sup}}$ | | 0.05 | 0.00 | 0.00 | 0.00 | 0.00 | 0.00 | 0.01 | 0.00 |
| $^{\dagger}L^{\mu}_{\mathrm{sub}}$ | | 0.01 | 0.00 | 0.00 | 0.00 | 1.03 | 0.30 | 0.01 | 0.00 |
| $L_{\mathrm{Info}}$ bound | Eq. (10) | 0.39 | 1.02 | 1.35 | 1.69 | 1.18 | 1.53 | 1.86 | 2.19 |
| $L^{\mu}_{\mathrm{sup}}$ | | 0.63 | 0.91 | 0.90 | 0.90 | 0.00 | 0.00 | 1.17 | 1.94 |
| $L^{\mu}_{\mathrm{sub}}$ | | 0.26 | 0.00 | 0.00 | 0.00 | 1.82 | 1.93 | 0.79 | 0.01 |
| Collision bound | Eq. (11) | 0.60 | 0.61 | 0.62 | 0.62 | 0.52 | 0.52 | 0.51 | 0.51 |

# 6 Related Work

## 6.1 Self-supervised Representation Learning

Self-supervised learning tries to learn an encoder that extracts generic feature representations from an unlabeled dataset. Self-supervised learning algorithms solve a pretext task that does not require any supervision and can be easily constructed on the dataset, such as denoising [Vincent et al., 2008], colorization [Zhang et al., 2016, Larsson et al., 2016] solving jigsaw puzzles [Noroozi and Favaro, 2016], inpainting blank pixels [Pathak et al., 2016], reconstructing missing channels [Zhang et al., 2017], predicting rotation [Gidaris et al., 2018], and adversarial generative models [Donahue and Simonyan, 2019] for vision; predicting neighbor words [Mikolov et al., 2013], generating neighbor sentences [Kiros et al., 2015], and solving masked language model [Devlin et al., 2019] for language. See also recent review articles [Le-Khac et al., 2020, Schmarje et al., 2021].

Recent self-supervised learning algorithms for the vision domain mainly solve an instance discrimination task. `Exemplar-CNN` [Dosovitskiy et al., 2014] is one of the earlier algorithms that can obtain generic feature representations of images by using convolutional neural networks and data augmentation. After van den Oord et al. [2018] proposed InfoNCE loss function, many state-of-the-art self-supervised algorithms minimize InfoNCE-based loss function, e.g., `DeepInfoMax` [Hjelm et al., 2019], `AMDIM` [Bachman et al., 2019], `SimCLR` [Chen et al., 2020a], `CPCv2` [Hénaff et al., 2020], `MoCo` [He et al., 2020], and `SwAV` [Caron et al., 2020].

## 6.2 Theoretical Perspective

InfoNCE [van den Oord et al., 2018] was initially proposed as a lower bound of intractable mutual information between feature representations by using noise-contrastive estimation (NCE) [Gutmann and Hyvärinen, 2012]. Optimizing InfoNCE can be considered as maximizing the InfoMax principle [Linsker, 1988]. The history of InfoMax-based self-supervised representation learning dates back to more than 30 years ago [Hinton and Becker, 1990]. However, this interpretation does not directly explain the generalization for a downstream task. Indeed, Tschannen et al. [2020] empirically showed that the performances of classification on downstream tasks and mutual information estimation are uncorrelated with each other. Kolesnikov et al. [2019] also reported a similar relationship between the performances of linear classification on downstream tasks and of pretext tasks. McAllester and Stratos [2020] theoretically showed the limitation of maximizing the lower bounds of mutual information – accurate approximation requires an exponential sample size.

As the most related work, Arora et al. [2019] provide the first theoretical analyses to explain the generalization of the CURL. It is worth noting that our analysis focuses on the different representation learning setting. Shortly after our publishing a draft of this paper on arXiv, Ash et al. [2021] also

published a paper on the role of negative samples in CURL bound with InfoNCE loss for fully supervised classification using the coupon collector's problem. Thus we believe that the coupon collector's problem is a key ingredient to analyze the connection between contrastive learning and supervised classification based on the CURL analysis by Arora et al. [2019]. Their work was developed independently of ours and their analysis is for contrastive unsupervised learning rather than self-supervised learning that is done in this work. The proposed bound by Ash et al. [2021] has also similar issue to Arora et al. [2019] as in our analysis; however, their bound holds with smaller $K$ than $C$. Bansal et al. [2021] decomposed the generalization error gap of a linear classifier given feature representations. Wang and Isola [2020] decomposed the self-supervised loss into alignment and uniformity and showed properties of both metrics. Li et al. [2021b] provided an interpretation of InfoNCE through a lens of kernel method. Mitrovic et al. [2021] provided another theoretical analysis with causality for instance discriminative self-supervised learning. Tosh et al. [2021] also analyzed self-supervised loss for augmented samples, but they only focused on one negative sample setting. Recently, Wei et al. [2021] proposed learning theoretical analysis by introducing "expansion" assumption for self-training where (pseudo) labels are generated from the previously learned model. Learning theory-based analyses were also proposed for other types of self-supervised learning problem such as reconstruction task [Garg and Liang, 2020, Lee et al., 2021] and language modeling [Saunshi et al., 2021]. The theoretical analysis on self-supervised representation algorithms without negative samples [Tian et al., 2021] cannot be applied to the contrastive learning setting.

### 6.3 Hard Negative Mining

In metric learning and contrastive learning, hard negative mining, such as Kalantidis et al. [2020], is actively proposed to make training more effective to avoid using inappropriate negative or too easy negative samples. The current work mainly focuses on the *quality* of negative samples rather than *quantity* of negative samples. However, removing false-negative samples can reduce the effect of the collision term in our bounds. Thus our analysis might provide a theoretical justification for hard negative mining.

## 7 Conclusion

We applied the CURL framework to the recent self-supervised representation learning formulation. We pointed out that the existing framework has difficulty explaining why large negative samples in self-supervised learning improve classification accuracy on a downstream supervised task as in Figure 1. We proposed a novel framework using the coupon collector's problem to explain the phenomenon and confirmed our analysis on real-world benchmark datasets.

**Limitations** We did not discuss the properties of data augmentation explicitly in our framework. Practically, self-supervised representation learning algorithms discard the projection head after self-supervised learning, but our analysis does not cover this procedure. We believe that extensions to cover these settings are fruitful explorations of future work.

### Acknowledgments

This work is supported (in part) by Next Generation AI Research Center, The University of Tokyo. The experiments were conducted using the SGI Rackable C2112-4GP3/C1102-GP8 (Reedbush-H/L) in the Information Technology Center, The University of Tokyo. We thank Han Bao, Yoshihiro Nagano, and Ikko Yamane for constructive discussion and Yusuke Tsuzuku for supporting our experiments. We also thank Junya Honda and Yivan Zhang for LaTeX support, specifically, for solving our font issue and MathJax, respectively. We appreciate anonymous reviewers of ICML 2021 and NeurIPS 2021 for giving constructive suggestions to improve our manuscript. KN is supported by JSPS KAKENHI Grant Number 18J20470.

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
