## A   Notations

Table 2 summarizes our notations.

## B   Relationship between Latent Classes and Supervised classes

In the main body of the manuscript, we assume that the latent classes $\mathcal{C}$ are a superset of the supervised classes $\mathcal{Y}$. Here, we mention a few scenarios covered by our analysis.

**Supervised Class is a Union of Latent Classes**   Suppose that latent classes $\mathcal{C}$ are breeds of dog and cat for image classification and supervised class $y \in \mathcal{Y}$ is a union of breeds: "dog" or "cat". In this setting, the downstream classifier can mispredict the class label among dog breeds for a dog image. Therefore the classifier's loss on all latent classes $\mathcal{Y} = \mathcal{C}$ is higher than or equal to the same classifier' loss on the union set at worst.

**Supervised Class is a Product of Latent Classes**   Another practical scenario is that the supervised class is a product of latent classes. Suppose that the latent classes are disentangled properties of dog/cat, such as breeds and color. In this case, a sample can be drawn from different latent classes, for example, a white golden retriever image can be drawn from "Golden Retriever" class and "white" class. Suppose that we perform classification over the product of the latent classes such as "White Golden Retriever" and "Black Ragamuffin" at the supervised step. Since we use Jensen's inequality to aggregate feature representation into the weights of mean classifier in Section 3.2, our analysis can deal with this case as well.

## C   Relation of Mean Classifier and Linear Classifier

Recall that $\mathbf{g} : \mathbb{R}^h \rightarrow \mathbb{R}^Y$ that is a function from a feature space to a label space. Suppose $\widehat{\mathbf{g}} = \operatorname{argmin}_{\mathbf{g}} L_{\mathrm{sup}}(\mathbf{g} \circ \mathbf{f})$. The mean classifier's supervised loss is bounded by a loss with $\widehat{\mathbf{g}}$:

$$L_{\mathrm{sup}}^{\mu}(\mathbf{f}) \geq L_{\mathrm{sup}}(\widehat{\mathbf{g}} \circ \mathbf{f}). \tag{12}$$

This relation holds with sub-class loss in Definition 4 as well.

Note that the mean classifier is a special case of the linear classifier [Snell et al., 2017, Sec. 2.4]. A linear classifier is defined as $\mathbf{W}\mathbf{f}(\mathbf{a}((\mathbf{x}))) + \mathbf{b}$, where $\mathbf{W} \in \mathbb{R}^{Y \times h}$ and $\mathbf{b} \in \mathbb{R}^Y$. When $\mathbf{W} = [\boldsymbol{\mu}_1, \ldots, \boldsymbol{\mu}_Y]^\top$ and $\mathbf{b} = \mathbf{0}$, the linear classifier is equivalent to the mean classifier.

## D   Proofs

### D.1   Inequality for Eq. (7)

We show the following inequality used to obtain Eq. (7):

$$\mathop{\mathbb{E}}_{c, \{c_k^-\}_{k=1}^K \sim \rho^{K+1}} \mathop{\mathbb{E}}_{\substack{\mathbf{x} \sim \mathcal{D}_c \\ \mathbf{a} \sim \mathcal{A}}} \ell_{\mathrm{Info}}\left(\mathbf{z}, \left\{\boldsymbol{\mu}(\mathbf{x}), \boldsymbol{\mu}_{c_1^-}, \ldots, \boldsymbol{\mu}_{c_K^-}\right\}\right)$$

$$\geq \mathop{\mathbb{E}}_{c, \{c_k^-\}_{k=1}^K \sim \rho^{K+1}} \mathop{\mathbb{E}}_{\substack{\mathbf{x} \sim \mathcal{D}_c \\ \mathbf{a} \sim \mathcal{A}}} \ell_{\mathrm{Info}}\left(\mathbf{z}, \left\{\boldsymbol{\mu}_c, \boldsymbol{\mu}_{c_1^-}, \ldots, \boldsymbol{\mu}_{c_K^-}\right\}\right) + d(\mathbf{f}). \tag{13}$$

Table 2: Notations

| Symbol | Description |
| --- | --- |
| $c$ | Latent class |
| $c^-$ | Latent class of a negative sample |
| $h$ | The dimensionality of feature representation $\mathbf{z}$ |
| $t$ | Temperature parameter in InfoNCE loss (1) |
| $y$ | Supervised class |
| $K$ | The number of negative samples |
| $N$ | The number of samples used in self-supervised learning |
| $N_y$ | The number of samples whose label is $y$ |
| $Y$ | The number of supervised classes |
| $\alpha$ | Non-negative coefficient in Eq. (11) |
| $\beta$ | Non-negative coefficient in Eq. (11) |
| $\tau$ | Collision probability in Proposition 6 |
| $\upsilon$ | Probability to draw all latent class defined in Section 4.1 |
| $\mathcal{C}$ | Latent classes associated with $\mathcal{Y}$ |
| $\mathcal{C}_{\mathrm{sub}}$ | Function to remove duplicated latent classes |
| $\mathcal{X}$ | Input space |
| $\mathcal{Y}$ | Supervised class space |
| $\mathcal{Y}_{\mathrm{sub}}$ | Subset of $\mathcal{Y}$ |
| $\mathcal{A}$ | Distribution of data augmentations |
| $\mathcal{D}_y$ | Distribution over $\mathcal{X}$ conditioned on supervised class $y$ |
| $\mathcal{D}_c$ | Distribution over $\mathcal{X}$ conditioned on latent class $c$ |
| $\mathcal{S}$ | Joint distribution over $\mathcal{X} \times \mathcal{Y}$ |
| $\mathcal{S}_{\mathrm{sub}}$ | Joint distribution over $\mathcal{X} \times \mathcal{Y}_{\mathrm{sub}}$ |
| $\rho$ | Probability distribution over $\mathcal{C}$ |
| $\mathbf{a}$ | Data augmentation: $\mathcal{X} \to \mathcal{X}$ |
| $\mathbf{a}^+$ | Data augmentation to create positive feature representation |
| $\mathbf{a}^-$ | Data augmentation to create negative feature representation |
| $\mathbf{f}$ | Feature extractor: $\mathcal{X} \to \mathbb{R}^h$ |
| $\widehat{\mathbf{f}}$ | Trained feature extractor by minimizing $\widehat{L}_{\mathrm{Info}}$ |
| $\mathbf{g}$ | Supervised classifier taken a feature representation $\mathbf{z}$: $\mathbb{R}^h \to \mathbb{R}^Y$ |
| $\mathbf{q}$ | Function that outputs real-valued vector used in Proposition 10 |
| $\mathbf{x}$ | Input sample in $\mathcal{X}$ |
| $\mathbf{z}$ | L2 normalized feature representation: $\mathbf{f}(\mathbf{a}(\mathbf{x}))$ |
| $\mathbf{z}^+$ | Positive L2 normalized feature representation: $\mathbf{f}(\mathbf{a}^+(\mathbf{x}))$ |
| $\mathbf{z}^-$ | Negative L2 normalized feature representation: $\mathbf{f}(\mathbf{a}^-(\mathbf{x}^-))$ |
| $\mathbf{Z}$ | Set of positive and $K$ negative features: $\{\mathbf{z}^+, \mathbf{z}_1^-, \ldots, \mathbf{z}_K^-\}$ |
| $\boldsymbol{\mu}(\mathbf{x})$ | Averaged feature representation over $\mathcal{A}$: $\mathbb{E}_{\mathbf{a} \sim \mathbf{A}} \mathbf{f}(\mathbf{a}(\mathbf{x}))$ |
| $\boldsymbol{\mu}_c$ | Mean classifier's weight vector of latent class $c$: $\mathbb{E}_{\mathbf{x} \sim \mathcal{D}_c} \boldsymbol{\mu}(\mathbf{x})$ |
| $\boldsymbol{\mu}_y$ | Mean classifier's weight vector of supervised class $y$: $\mathbb{E}_{\mathbf{x} \sim \mathcal{D}_y} \boldsymbol{\mu}(\mathbf{x})$ |
| $\mathrm{Col}(\cdot, \cdot)$ | Collision value defined in Section 3.2: $\sum_{k=1}^K \mathbb{I}[c^k = c_k^-]$ |
| $d(\cdot)$ | Gap term defined in Eq. (7) |
| $\ell_{\mathrm{Info}}$ | InfoNCE-based self-supervised loss defined by Eq. (1) |
| $\ell_{\mathrm{sub}}^\mu$ | Mean classifier's supervised sub-class loss defined by Eq. (16) |
| $L_{\mathrm{Info}}$ | Expected self-supervised loss defined by Eq. (2) |
| $\widehat{L}_{\mathrm{Info}}$ | Empirical self-supervised loss |
| $L_{\mathrm{sup}}$ | Supervised loss defined by Eq. (3) |
| $L_{\mathrm{sub}}$ | Supervised sub-class loss defined by Eq. (5) |
| $L_{\mathrm{sup}}^\mu$ | Supervised loss of mean classifier defined by Eq. (4) |
| $L_{\mathrm{sub}}^\mu$ | Sub-class supervised loss of mean classifier by Eq. (6) |
| $\mathbb{I}[\cdot]$ | Indicator function |
| $(\cdot \cdot)$ | Inner product |

*Proof.* We replace $\boldsymbol{\mu}(\mathbf{x})$ with $\boldsymbol{\mu}_c$ in the left hand side of Eq. (13):

$$\mathbb{E}_{c,\{c_k^-\}_{k=1}^K \sim \rho^{K+1}} \quad \mathbb{E}_{\substack{\mathbf{x} \sim \mathcal{D}_c \\ \mathbf{a} \sim \mathcal{A}}} \ell_{\text{Info}}\left(\mathbf{z}, \left\{\boldsymbol{\mu}(\mathbf{x}), \boldsymbol{\mu}_{c_1^-}, \ldots, \boldsymbol{\mu}_{c_K^-}\right\}\right)$$

$$= \mathbb{E}_{c,\{c_k^-\}_{k=1}^K \sim \rho^{K+1}} \quad \mathbb{E}_{\substack{\mathbf{x} \sim \mathcal{D}_c \\ \mathbf{a} \sim \mathcal{A}}} \ell_{\text{Info}}\left(\mathbf{z}, \left\{\boldsymbol{\mu}(\mathbf{x}), \boldsymbol{\mu}_{c_1^-}, \ldots, \boldsymbol{\mu}_{c_K^-}\right\}\right)$$

$$+ \mathbb{E}_{c,\{c_k^-\}_{k=1}^K \sim \rho^{K+1}} \quad \mathbb{E}_{\substack{\mathbf{x} \sim \mathcal{D}_c \\ \mathbf{a} \sim \mathcal{A}}} \ell_{\text{Info}}\left(\mathbf{z}, \left\{\boldsymbol{\mu}_c, \boldsymbol{\mu}_{c_1^-}, \ldots, \boldsymbol{\mu}_{c_K^-}\right\}\right)$$

$$- \mathbb{E}_{c,\{c_k^-\}_{k=1}^K \sim \rho^{K+1}} \quad \mathbb{E}_{\substack{\mathbf{x} \sim \mathcal{D}_c \\ \mathbf{a} \sim \mathcal{A}}} \ell_{\text{Info}}\left(\mathbf{z}, \left\{\boldsymbol{\mu}_c, \boldsymbol{\mu}_{c_1^-}, \ldots, \boldsymbol{\mu}_{c_K^-}\right\}\right)$$

$$= \mathbb{E}_{c,\{c_k^-\}_{k=1}^K \sim \rho^{K+1}} \quad \mathbb{E}_{\substack{\mathbf{x} \sim \mathcal{D}_c \\ \mathbf{a} \sim \mathcal{A}}} \ell_{\text{Info}}\left(\mathbf{z}, \left\{\boldsymbol{\mu}_c, \boldsymbol{\mu}_{c_1^-}, \ldots, \boldsymbol{\mu}_{c_K^-}\right\}\right)$$

$$+ \underbrace{\mathbb{E}_{c,\{c_k^-\}_{k=1}^K \sim \rho^{K+1}} \quad \mathbb{E}_{\substack{\mathbf{x} \sim \mathcal{D}_c \\ \mathbf{a} \sim \mathcal{A}}} \left[\ell_{\text{Info}}\left(\mathbf{z}, \left\{\boldsymbol{\mu}(\mathbf{x}), \boldsymbol{\mu}_{c_1^-}, \ldots, \boldsymbol{\mu}_{c_K^-}\right\}\right) - \ell_{\text{Info}}\left(\mathbf{z}, \left\{\boldsymbol{\mu}_c, \boldsymbol{\mu}_{c_1^-}, \ldots, \boldsymbol{\mu}_{c_K^-}\right\}\right)\right]}_{\text{Gap term}}.$$

$$(14)$$

When the gap term is non-negative: $\mathbf{z} \cdot \boldsymbol{\mu}(\mathbf{x}) \leq \mathbf{z} \cdot \boldsymbol{\mu}_c$, we drop it from Eq. (14) to obtain the lower bound (13) with $d(\mathbf{f}) = 0$. Thus we consider a lower bound of the gap term when the gap term is negative, $\mathbf{z} \cdot \boldsymbol{\mu}(\mathbf{x}) > \mathbf{z} \cdot \boldsymbol{\mu}_c$, to guarantee Eq. (14). As a general case, we show the bound with temperature parameter $t$.

The gap term in Eq. (14)

$$= \mathbb{E}_{c,\{c_k^-\}_{k=1}^K \sim \rho^{K+1}} \quad \mathbb{E}_{\substack{\mathbf{x} \sim \mathcal{D}_c \\ \mathbf{a} \sim \mathcal{A}}} [\mathbf{z} \cdot (\boldsymbol{\mu}_c - \boldsymbol{\mu}(\mathbf{x}))/t] + \underbrace{\ln \frac{\exp\left(\mathbf{z} \cdot \boldsymbol{\mu}(\mathbf{x})/t\right) + \sum_{k=1}^K \exp\left(\mathbf{z} \cdot \boldsymbol{\mu}_{c_k^-}/t\right)}{\exp(\mathbf{z} \cdot \boldsymbol{\mu}_c/t) + \sum_{k=1}^K \exp\left(\mathbf{z} \cdot \boldsymbol{\mu}_{c_k^-}/t\right)}}_{\text{Non-negative}}$$

$$> \frac{1}{t} \mathbb{E}_{c \sim \rho} \mathbb{E}_{\substack{\mathbf{x} \sim \mathcal{D}_c \\ \mathbf{a} \sim \mathcal{A}}} [\mathbf{z} \cdot (\boldsymbol{\mu}_c - \boldsymbol{\mu}(\mathbf{x}))]$$

$$= \frac{1}{t} \mathbb{E}_{c \sim \rho} \mathbb{E}_{\mathbf{x} \sim \mathcal{D}_c} \left[\mathbb{E}_{\mathbf{a} \sim \mathcal{A}} [\mathbf{f}(\mathbf{a}(\mathbf{x}))] \cdot \left(\mathbb{E}_{\substack{\mathbf{x}^+ \sim \mathcal{D}_c \\ \mathbf{a}_1^+ \sim \mathcal{A}}} [\mathbf{f}(\mathbf{a}_1^+(\mathbf{x}^+))] - \mathbb{E}_{\mathbf{a}_2^+ \sim \mathcal{A}} [\mathbf{f}(\mathbf{a}_2^+(\mathbf{x}))]\right)\right] \qquad (15)$$

$$:= d(\mathbf{f}).$$

$\square$

## D.2  Proof of Proposition 6

**Proposition 6** (CURL Lower Bound of Self-supervised Loss). *For all feature extractor* $\mathbf{f}$,

$$L_{\text{Info}}(\mathbf{f}) \geq (1 - \tau_K) \mathbb{E}_{c,\{c_k^-\}_{k=1}^K \sim \rho^{K+1}} [\underbrace{L_{\text{sub}}^{\mu}(\mathbf{f}, \mathcal{C}_{\text{sub}})}_{\textit{sub-class loss}} \mid \text{Col} = 0]$$

$$+ \tau_K \mathbb{E}_{c,\{c_k^-\}_{k=1}^K \sim \rho^{K+1}} [\underbrace{\ln(\text{Col} + 1)}_{\textit{collision}} \mid \text{Col} \neq 0] + d(\mathbf{f}). \qquad (8)$$

We apply the proof of Theorem B.1 in Arora et al. [2019] to the self-supervised learning setting.

*Proof.* Before showing the inequality, we define the following sub-class loss with mean classifier for single $\mathbf{z}$ with $c$ and $\mathcal{C}_{\text{sub}}(\{c, c_1^-, \ldots, c_K^-\})$:

$$\ell_{\text{sub}}^{\mu}(\mathbf{z}, c, \mathcal{C}_{\text{sub}}) := -\ln \frac{\exp\left(\mathbf{z} \cdot \boldsymbol{\mu}_c\right)}{\sum_{j \in \mathcal{C}_{\text{sub}}} \exp\left(\mathbf{z} \cdot \boldsymbol{\mu}_j\right)}. \qquad (16)$$

We start from Eq. (7).

$$
\begin{aligned}
(7) = (1 - \tau_K) &\underset{c,\{c_k^-\}_{k=1}^K \sim \rho^{K+1}}{\mathbb{E}} \underset{\substack{\mathbf{x} \sim \mathcal{D}_c \\ \mathbf{a} \sim \mathcal{A}}}{\mathbb{E}} \left[ \ell_{\mathrm{Info}}\left( \mathbf{z}, \left\{ \boldsymbol{\mu}_c, \boldsymbol{\mu}_{c_1^-}, \ldots, \boldsymbol{\mu}_{c_K^-} \right\} \right) \,\middle|\, \mathrm{Col}\left( c, \{c_k^-\}_{k=1}^K \right) = 0 \right] \\
+ \tau_K &\underset{c,\{c_k^-\}_{k=1}^K \sim \rho^{K+1}}{\mathbb{E}} \underset{\substack{\mathbf{x} \sim \mathcal{D}_c \\ \mathbf{a} \sim \mathcal{A}}}{\mathbb{E}} \left[ \ell_{\mathrm{Info}}\left( \mathbf{z}, \left\{ \boldsymbol{\mu}_c, \boldsymbol{\mu}_{c_1^-}, \ldots, \boldsymbol{\mu}_{c_K^-} \right\} \right) \,\middle|\, \mathrm{Col}\left( c, \{c_k^-\}_{k=1}^K \right) \neq 0 \right] + d(\mathbf{f}) \\
\geq (1 - \tau_K) &\underset{c,\{c_k^-\}_{k=1}^K \sim \rho^{K+1}}{\mathbb{E}} \underset{\substack{\mathbf{x} \sim \mathcal{D}_c \\ \mathbf{a} \sim \mathcal{A}}}{\mathbb{E}} \left[ \ell_{\mathrm{sub}}^{\mu}(\mathbf{z}, c, \mathcal{C}_{\mathrm{sub}}) \,\middle|\, \mathrm{Col} = 0 \right] \\
+ \tau_K &\underset{c,\{c_k^-\}_{k=1}^K \sim \rho^{K+1}}{\mathbb{E}} \underset{\substack{\mathbf{x} \sim \mathcal{D}_c \\ \mathbf{a} \sim \mathcal{A}}}{\mathbb{E}} \left[ \ell_{\mathrm{sub}}^{\mu}(\mathbf{z}, c, \{c\}^{\mathrm{Col}+1}) \,\middle|\, \mathrm{Col} \neq 0 \right] + d(\mathbf{f}) \\
= (1 - \tau_K) &\underset{c,\{c_k^-\}_{k=1}^K \sim \rho^{K+1}}{\mathbb{E}} \left[ L_{\mathrm{sub}}^{\mu}(\mathbf{z}, \mathcal{C}_{\mathrm{sub}}) \,\middle|\, \mathrm{Col} = 0 \right] \\
+ \tau_K &\underset{c,\{c_k^-\}_{k=1}^K \sim \rho^{K+1}}{\mathbb{E}} \left[ \ln(\mathrm{Col} + 1) \,\middle|\, \mathrm{Col} \neq 0 \right] + d(\mathbf{f}), \quad (17)
\end{aligned}
$$

where $\{c\}^{\mathrm{Col}+1}$ is a bag of latent classes that contains only $c$ and the number of $c$ is $\mathrm{Col}(c, \{c_k^-\}_{k=1}^K) + 1$.

The first equation is done by using collision probability $\tau_K$ conditioned on $\mathrm{Col}$. The inequality is obtained by removing duplicated latent classes from the first term and removing the other latent classes from the second term. Note that InfoNCE is a monotonically increasing function; its value decreases by removing any elements in the negative features in the loss. □

### D.3 Proof of Theorem 8

**Theorem 8** (Proposed Lower Bound of Self-supervised Loss). *For all feature extractor $\mathbf{f}$,*

$$
\begin{aligned}
L_{\mathrm{Info}}(\mathbf{f}) \geq \frac{1}{2} \Big\{ &\upsilon_{K+1} \underset{c,\{c_k^-\}_{k=1}^K \sim \rho^{K+1}}{\mathbb{E}} \underbrace{\left[ L_{\mathrm{sub}}^{\mu}(\mathbf{f}, \mathcal{C}) \,\middle|\, \mathcal{C}_{\mathrm{sub}} = \mathcal{C} \right]}_{\textit{sup. loss}} \\
+ &(1 - \upsilon_{K+1}) \underset{c,\{c_k^-\}_{k=1}^K \sim \rho^{K+1}}{\mathbb{E}} \underbrace{\left[ L_{\mathrm{sub}}^{\mu}(\mathbf{f}, \mathcal{C}_{\mathrm{sub}}) \,\middle|\, \mathcal{C}_{\mathrm{sub}} \neq \mathcal{C} \right]}_{\textit{sub-class loss}} \\
+ &\underset{c,\{c_k^-\}_{k=1}^K \sim \rho^{K+1}}{\mathbb{E}} \underbrace{\ln(\mathrm{Col} + 1)}_{\textit{collision}} \Big\} + d(\mathbf{f}). \quad (10)
\end{aligned}
$$

*Proof.* We start from Eq. (7).

$$
\begin{aligned}
(7) = \ \ &\upsilon_{K+1} \underset{c,\{c_k^-\}_{k=1}^K \sim \rho^{K+1}}{\mathbb{E}} \underset{\substack{\mathbf{x} \sim \mathcal{D}_c \\ \mathbf{a} \sim \mathcal{A}}}{\mathbb{E}} \left[ \ell_{\mathrm{Info}}\left( \mathbf{z}, \left\{ \boldsymbol{\mu}_c, \boldsymbol{\mu}_{c_1^-}, \ldots, \boldsymbol{\mu}_{c_K^-} \right\} \right) \,\middle|\, \mathcal{C}_{\mathrm{sub}}(\{c, c_1^-, \ldots, c_K^-\}) = \mathcal{C} \right] \\
+ &(1 - \upsilon_{K+1}) \underset{c,\{c_k^-\}_{k=1}^K \sim \rho^{K+1}}{\mathbb{E}} \underset{\substack{\mathbf{x} \sim \mathcal{D}_c \\ \mathbf{a} \sim \mathcal{A}}}{\mathbb{E}} \left[ \ell_{\mathrm{Info}}\left( \mathbf{z}, \left\{ \boldsymbol{\mu}_c, \boldsymbol{\mu}_{c_1^-}, \ldots, \boldsymbol{\mu}_{c_K^-} \right\} \right) \,\middle|\, \mathcal{C}_{\mathrm{sub}}(\{c, c_1^-, \ldots, c_K^-\}) \neq \mathcal{C} \right] + d(\mathbf{f})
\end{aligned}
$$
$$(18)$$

$$
\begin{aligned}
\geq \ \ \frac{1}{2} \Big\{ &\upsilon_{K+1} \underset{c,\{c_k^-\}_{k=1}^K \sim \rho^{K+1}}{\mathbb{E}} \underset{\substack{\mathbf{x} \sim \mathcal{D}_c \\ \mathbf{a} \sim \mathcal{A}}}{\mathbb{E}} \left[ \ell_{\mathrm{sub}}^{\mu}(\mathbf{z}, c, \mathcal{C}_{\mathrm{sub}}) \,\middle|\, \mathcal{C}_{\mathrm{sub}} = \mathcal{C} \right] \\
+ &(1 - \upsilon_{K+1}) \underset{c,\{c_k^-\}_{k=1}^K \sim \rho^{K+1}}{\mathbb{E}} \underset{\substack{\mathbf{x} \sim \mathcal{D}_c \\ \mathbf{a} \sim \mathcal{A}}}{\mathbb{E}} \left[ \ell_{\mathrm{sub}}^{\mu}(\mathbf{z}, c, \mathcal{C}_{\mathrm{sub}}) \,\middle|\, \mathcal{C}_{\mathrm{sub}} \neq \mathcal{C} \right] \\
+ &\underset{c,\{c_k^-\}_{k=1}^K \sim \rho^{K+1}}{\mathbb{E}} \ln\left( \mathrm{Col}\left( c, \{c_k^-\}_{k=1}^K \right) + 1 \right) \Big\} + d(\mathbf{f}), \quad (19)
\end{aligned}
$$

where recall that $\ell_{\mathrm{sub}}^{\mu}$ is defined by Eq. (16).

The first equation is obtained by using the definition of $\upsilon$ to distinguish whether or not the sampled latent classes contain all latent classes $\mathcal{C}$. To derive the inequality, we use the following properties of

the cross-entropy loss. Fixed $\mathbf{f}$ and $c$, for all $\mathcal{K} \subseteq \{1, \ldots, K\}$, the value of $\ell_{\text{Info}}$ holds the following inequality:

$$\ell_{\text{Info}}\left(\mathbf{z}, \left\{\{\boldsymbol{\mu}_c\} \cup \{\boldsymbol{\mu}_{c_k^-}\}_{k=1}^K\right\}\right) \geq \ell_{\text{Info}}\left(\mathbf{z}, \{\boldsymbol{\mu}_c\} \cup \{\boldsymbol{\mu}_{c_k^-}\}_{k \in \mathcal{K}}\right). \tag{20}$$

Thus we apply the following inequalities to the first and second terms in Eq. (18).

$$\ell_{\text{Info}}\left(\mathbf{z}, \left\{\boldsymbol{\mu}_c, \boldsymbol{\mu}_{c_1^-}, \ldots, \boldsymbol{\mu}_{c_K^-}\right\}\right) \geq \frac{1}{2}\left[\ell_{\text{sub}}^\mu\left(\mathbf{z}, c, \mathcal{C}_{\text{sub}}\right) + \ell_{\text{sub}}^\mu\left(\mathbf{z}, c, \{c\}^{\text{Col}+1}\right)\right]$$

$$= \frac{1}{2}\left[\ell_{\text{sub}}^\mu\left(\mathbf{z}, c, \mathcal{C}_{\text{sub}}\right) + \ln(\text{Col}+1)\right] \tag{21}$$

By following the notations of $L_{\text{sup}}^\mu$ and $L_{\text{sub}}^\mu$, we obtain the lower bound (10) in Theorem 8 from Eq. (19). $\qquad\square$

### D.4 Proof of Corollary 9

**Corollary 9** (Upper Bound of Collision Term). *Given a latent class $c$, $K$ negative classes $\{c_k^-\}_{k=1}^K$, and feature extractor $\mathbf{f}$,*

$$\ln\left(\text{Col}\left(c, \{c_k^-\}_{k=1}^K\right) + 1\right) \leq \alpha + \beta \underset{\substack{\mathbf{x} \sim \mathcal{D}_c \\ (\mathbf{a}, \mathbf{a}^+) \sim \mathcal{A}^2}}{\mathbb{E}} \underset{\substack{\mathbf{x}' \sim \mathcal{D}_c \\ \mathbf{a}' \sim \mathcal{A}}}{\mathbb{E}} \left|\mathbf{f}(\mathbf{a}(\mathbf{x})) \cdot \left[\mathbf{f}(\mathbf{a}'(\mathbf{x}')) - \mathbf{f}(\mathbf{a}^+(\mathbf{x}))\right]\right|, \tag{11}$$

*where $\alpha$ and $\beta$ are non-negative constants depending on the number of duplicated latent classes.*

We prove Corollary 9 that shows the upper bound of the collision term to understand the effect of the number of negative samples. Our proof is inspired by Arora et al. [2019, Lemma A.1]. As a general case, we consider the loss function with temperature parameter $t$.

*Proof.* We decompose the self-supervised loss $L_{\text{Info}}$ by using $\upsilon$ to extract the collision term.

$L_{\text{Info}}(\mathbf{f})$

$$= \upsilon_{K+1}\left[\underset{\substack{c,\{c_k^-\}_{k=1}^K \sim \rho^{K+1} \\ (\mathbf{a}, \mathbf{a}^+) \sim \mathcal{A}^2}}{\mathbb{E}} \underset{\substack{\mathbf{x} \sim \mathcal{D}_c \\ \{\mathbf{a}_k^-\}_{k=1}^K \sim \mathcal{A}^K}}{\mathbb{E}} \underset{\{\mathbf{x}_k^- \sim \mathcal{D}_{c_k^-}\}_{k=1}^K}{\mathbb{E}} \left[\ell_{\text{Info}}(\mathbf{z}, \mathbf{Z}) \mid \mathcal{C}_{\text{sub}}(\{c, c_1^-, \ldots, c_K^-\}) = \mathcal{C}\right]\right]$$

$$+ (1 - \upsilon_{K+1})\left[\underset{\substack{c,\{c_k^-\}_{k=1}^K \sim \rho^{K+1} \\ (\mathbf{a}, \mathbf{a}^+) \sim \mathcal{A}^2}}{\mathbb{E}} \underset{\substack{\mathbf{x} \sim \mathcal{D}_c \\ \{\mathbf{a}_k^-\}_{k=1}^K \sim \mathcal{A}^K}}{\mathbb{E}} \underset{\{\mathbf{x}_k^- \sim \mathcal{D}_{c_k^-}\}_{k=1}^K}{\mathbb{E}} \left[\ell_{\text{Info}}(\mathbf{z}, \mathbf{Z}) \mid \mathcal{C}_{\text{sub}}(\{c, c_1^-, \ldots, c_K^-\}) \neq \mathcal{C}\right]\right]$$

$$\leq \upsilon_{K+1}\left[\underset{\substack{c,\{c_k^-\}_{k=1}^K \sim \rho^{K+1} \\ (\mathbf{a}, \mathbf{a}^+) \sim \mathcal{A}^2}}{\mathbb{E}} \underset{\substack{\mathbf{x} \sim \mathcal{D}_c \\ \{\mathbf{a}_k^-\}_{k=1}^K \sim \mathcal{A}^K}}{\mathbb{E}} \underset{\{\mathbf{x}_k^- \sim \mathcal{D}_{c_k^-}\}_{k=1}^K}{\mathbb{E}} \left[\ell_{\text{Info}}\left(\mathbf{z}, \{\mathbf{z}_k\}_{c \neq c_k^-}\right) + \ell_{\text{Info}}\left(\mathbf{z}, \{\mathbf{z}_k\}_{c = c_k^-}\right) \mid \mathcal{C}_{\text{sub}} = \mathcal{C}\right]\right]$$

$$+ (1 - \upsilon_{K+1})\left[\underset{\substack{c,\{c_k^-\}_{k=1}^K \sim \rho^{K+1} \\ (\mathbf{a}, \mathbf{a}^+) \sim \mathcal{A}^2}}{\mathbb{E}} \underset{\substack{\mathbf{x} \sim \mathcal{D}_c \\ \{\mathbf{a}_k^-\}_{k=1}^K \sim \mathcal{A}^K}}{\mathbb{E}} \underset{\{\mathbf{x}_k^- \sim \mathcal{D}_{c_k^-}\}_{k=1}^K}{\mathbb{E}} \left[\ell_{\text{Info}}\left(\mathbf{z}, \{\mathbf{z}_k\}_{c \neq c_k^-}\right) + \ell_{\text{Info}}\left(\mathbf{z}, \{\mathbf{z}_k\}_{c = c_k^-}\right) \mid \mathcal{C}_{\text{sub}} \neq \mathcal{C}\right]\right]$$

$$= \underset{\substack{c,\{c_k^-\}_{k=1}^K \sim \rho^{K+1} \\ (\mathbf{a}, \mathbf{a}^+) \sim \mathcal{A}^2}}{\mathbb{E}} \underset{\substack{\mathbf{x} \sim \mathcal{D}_c \\ \{\mathbf{a}_k^-\}_{k=1}^K \sim \mathcal{A}^K}}{\mathbb{E}} \underset{\{\mathbf{x}_k^- \sim \mathcal{D}_{c_k^-}\}_{k=1}^K}{\mathbb{E}} \ell_{\text{Info}}\left(\mathbf{z}, \{\mathbf{z}_k\}_{c = c_k^-}\right) + \text{Reminder}$$

$$= \underset{c \sim \rho \ \text{Col} \sim \mathcal{B}_c}{\mathbb{E}} \underset{\substack{\mathbf{x}, \{\mathbf{x}_k^-\}_{k=1}^{\text{Col}} \sim \mathcal{D}_c^{\text{Col}+1} \\ (\mathbf{a}, \mathbf{a}^+, \{\mathbf{a}_k^-\}_{k=1}^{\text{Col}}) \sim \mathcal{A}^{\text{Col}+2}}}{\mathbb{E}} \ell_{\text{Info}}(\mathbf{z}, \mathbf{Z}) + \text{Reminder}, \tag{22}$$

where $\mathcal{B}_c$ is the probability distribution over Col conditioned on $c$ with $K$. The first equality is done by decomposition with $\upsilon$. The inequality is obtained by using the following property of the cross-entropy loss. Given $\mathcal{K} \subseteq \{1, \ldots, K\}$,

$$\ell_{\text{Info}}(\mathbf{z}, \mathbf{Z}) \leq \ell_{\text{Info}}\left(\mathbf{z}, \{\mathbf{z}^+\} \cup \{\mathbf{z}_k^-\}_{k \in \mathcal{K}}\right) + \ell_{\text{Info}}\left(\mathbf{z}, \{\mathbf{z}^+\} \cup \{\mathbf{z}_k^-\}_{k \in \{1, \ldots, K\} \setminus \mathcal{K}}\right). \qquad (23)$$

We focus on the first term in Eq. (22), where the loss takes samples are drawn from the same latent class $c$. Fixed $\mathbf{f}$ and $c$, let $m = \max_{\mathbf{z}_k \in \mathbf{Z}} \mathbf{z} \cdot \mathbf{z}_k / t$ and $\mathbf{z}^* = \operatorname{argmax}_{\mathbf{z}_k \in \mathbf{Z}} \mathbf{z} \cdot \mathbf{z}_k$.

$$\mathbb{E}_{\substack{c \sim \rho \\ \text{Col} \sim \mathcal{B}_c \\ \mathbf{x}, \{\mathbf{x}_k^-\}_{k=1}^{\text{Col}} \sim \mathcal{D}_c^{\text{Col}+1} \\ (\mathbf{a}, \mathbf{a}^+, \{\mathbf{a}_k^-\}_{k=1}^{\text{Col}}) \sim \mathcal{A}^{\text{Col}+2}}} \ell_{\text{Info}}(\mathbf{z}, \mathbf{Z})$$

$$\leq \mathbb{E}_{\substack{c \sim \rho \\ \text{Col} \sim \mathcal{B}_c \\ \mathbf{x}, \{\mathbf{x}_k^-\}_{k=1}^{\text{Col}} \sim \mathcal{D}_c^{\text{Col}+1} \\ (\mathbf{a}, \mathbf{a}^+, \{\mathbf{a}_k^-\}_{k=1}^{\text{Col}}) \sim \mathcal{A}^{\text{Col}+2}}} -\mathbf{z} \cdot \mathbf{z}^+ / t + \ln\left[\exp(\mathbf{z} \cdot \mathbf{z}^+ / t) + \text{Col} \exp(m)\right]$$

$$\leq \mathbb{E}_{\substack{c \sim \rho \\ \text{Col} \sim \mathcal{B}_c \\ \mathbf{x}, \{\mathbf{x}_k^-\}_{k=1}^{\text{Col}} \sim \mathcal{D}_c^{\text{Col}+1} \\ (\mathbf{a}, \mathbf{a}^+, \{\mathbf{a}_k^-\}_{k=1}^{\text{Col}}) \sim \mathcal{A}^{\text{Col}+2}}} \max\left[\ln(\text{Col} + 1), \ln(\text{Col} + 1) - \mathbf{z} \cdot \mathbf{z}^+ / t + m\right]$$

$$\leq \mathbb{E}_{\substack{c \sim \rho \\ \text{Col} \sim \mathcal{B}_c \\ \mathbf{x}, \{\mathbf{x}_k^-\}_{k=1}^{\text{Col}} \sim \mathcal{D}_c^{\text{Col}+1} \\ (\mathbf{a}, \mathbf{a}^+, \{\mathbf{a}_k^-\}_{k=1}^{\text{Col}}) \sim \mathcal{A}^{\text{Col}+2}}} \ln(\text{Col} + 1) + \frac{1}{t}|\mathbf{z} \cdot \mathbf{z}^* - \mathbf{z} \cdot \mathbf{z}^+|. \qquad (24)$$

Note that $\ln(\text{Col} + 1)$ is the constant depending on the number of duplicated latent classes, then we focus on $|\mathbf{z} \cdot \mathbf{z}^* - \mathbf{z} \cdot \mathbf{z}^+|$.

$$\mathbb{E}_{\substack{\mathbf{x}, \{\mathbf{x}_k^-\}_{k=1}^{\text{Col}} \sim \mathcal{D}_c^{\text{Col}+1} \\ (\mathbf{a}, \mathbf{a}^+, \{\mathbf{a}_k^-\}_{k=1}^{\text{Col}}) \sim \mathcal{A}^{\text{Col}+2}}} |\mathbf{z} \cdot \mathbf{z}^* - \mathbf{z} \cdot \mathbf{z}^+| \leq (\text{Col} + 1) \mathbb{E}_{\substack{\mathbf{x} \sim \mathcal{D}_c \\ (\mathbf{a}, \mathbf{a}^+) \sim \mathcal{A}^2}} \mathbb{E}_{\substack{\mathbf{x}' \sim \mathcal{D}_c \\ \mathbf{a}' \sim \mathcal{A}}} |\mathbf{z} \cdot \mathbf{z}' - \mathbf{z} \cdot \mathbf{z}^+|.$$

Therefore, $\alpha = \ln(\text{Col} + 1)$ and $\beta = \frac{\text{Col}+1}{t}$. $\qquad \square$

### D.5 Proof of Proposition 10

**Proposition 10** (Optimality of $L_{\text{sub}}$). *Suppose $\mathcal{C} \supseteq \mathcal{Y}$ and $\rho$ is uniform: $\forall c \in \mathcal{C}, \rho(c) = 1/|\mathcal{C}|$. Suppose a constant function $\mathbf{q} : \mathbf{x} \in \mathcal{X} \overset{\mathbf{q}}{\mapsto} [q_1, \ldots, q_{|\mathcal{C}|}]^\top$. Optimal $\mathbf{q}^*$ is a constant function that outputs a vector with the same value if and only if it minimizes $\mathbb{E}_{c, \{c_k^-\}_{k=1}^K \sim \rho^{K+1}} L_{\text{sub}}(\mathbf{q}^*, \mathcal{C}_{\text{sub}})$. The optimal $\mathbf{q}^*$ is also the minimizer of $L_{\text{sup}}$.*

*Proof.* Suppose an observed set of data for supervised sub-class loss $\{(\mathbf{x}_i, \mathcal{C}_{\text{sub},i}(\{c_i, c_1^-, \ldots, c_K^-\}))\}_{i=1}^M$, where $\mathcal{C}_{\text{sub},i}$ is a subset of $\mathcal{C}$ such that $\mathcal{C}_{\text{sub},i}$ contains $c_i$ of $\mathbf{x}_i$ and it holds $2 \leq |\mathcal{C}_{\text{sub},i}| \leq K + 1$. We take derivatives of the empirical sub-class loss $\frac{1}{M} \sum_{i=1}^M -\ln \frac{\exp(q_{c_i})}{\sum_{j \in \mathcal{C}_{\text{sub},i}} \exp(q_j)}$ with respect to each element of $\mathbf{q}$, then the stationary points are $\forall c \in \mathcal{C}$,

$$\sum_{n_1 \in \mathcal{C} \setminus c} N_{c,n_1} \left(1 - 2\frac{\exp(q_c)}{\sum_{j \in \{c,n_1\}} \exp(q_j)}\right)$$

$$+ \sum_{n_1, n_2 \in \mathcal{C} \setminus c} N_{c,n_1,n_2} \left(1 - 3\frac{\exp(q_c)}{\sum_{j \in \{c,n_1,n_2\}} \exp(q_j)}\right)$$

$$\vdots$$

$$+ \sum_{n_1, \ldots, n_K \in \mathcal{C} \setminus c} N_{c,n_1,\ldots,n_K} \left(1 - (K+1)\frac{\exp(q_c)}{\sum_{j \in \{c,n_1,\ldots,n_K\}} \exp(q_j)}\right) = 0, \qquad (25)$$

Table 3: The expected number of samples to draw all supervised classes.

| Dataset | # classes | $\mathbb{E}[K+1]$ |
|---|---|---|
| AGNews | 4 | 9 |
| CIFAR-10 | 10 | 30 |
| CIFAR-100 | 100 | 519 |
| ImageNet | 1 000 | 7 709 |

where $N_{c,n_1,\ldots,n_K}$ is the frequency of $\mathcal{C}_{\text{sub},i}$ such that $\sum_{i=1}^M \mathbb{I}[\mathcal{C}_{\text{sub},i} = \{c_i\} \cup \{n_1, \ldots, n_K\}]$. As a result, the optimal $\mathbf{q}^*$ is a constant function with the same value. For supervised loss defined in Eq. (3), the optimal score function of class $y$ is $q_y = \ln N_y + \text{Constant}$, where $N_y$ is the number of samples whose label is $y$ and $\text{Constant} \in \mathbb{R}$. From the uniform assumption, the optimal $\mathbf{q}^*$ is the minimizer of $L_{\text{sup}}$. $\qquad\square$

## E  Expected Number of Negative Samples to Draw all Supervised Labels

We assume that we sample a latent class from $\rho$ independently. Let $\rho(c) \in [0, 1]$ be a probability that $c$ is sampled. Flajolet et al. [1992, Theorem 4.1] show the expected value of $K + 1$ to sample all latent class in $\mathcal{C}$ is defined by

$$\mathbb{E}[K+1] = \int_0^\infty \left( 1 - \prod_{c \in \mathcal{C}} [1 - \exp(-\rho(c)x)] \right) \mathrm{d}x. \tag{26}$$

Table 3 shows the expected number of sampled latent classes on a popular classification dataset when the latent class is the same as the supervised class. For ImageNet, we use the relative frequency of supervised classes in the training dataset as $\rho$. According to Table 3, the empirical number of negative samples is supposed to be natural, for example, $K \geq 8\,096$ in experiments by Chen et al. [2020a], He et al. [2020].

## F  Additional Experimental Results

### F.1  Experimental Results related to Upper Bound of Collision Term in Corollary 9

As shown in Table 1, the upper bound values of the collision term did not increase by increasing $K$. We further analyzed representations in terms of the number of negative samples $K$.

As we discuss in the Section 4.2, we expect that cosine similar values between samples in the same latent class do not change by increasing $K$. To confirm that, Figures 2 and 3 show histgrams of cosine similarity values between all pairs of feature representations in the same supervised class. The cosine similarity values do not change by increasing $K$ in practice. We used feature representation $\widehat{\mathbf{f}}(\mathbf{x})$ on the training dataset. We used the number of bins of each histogram as the square root of the number of cosine similarity values in each class. We did not use duplicated similarity values: similarity between $\widehat{\mathbf{f}}(\mathbf{x}_i)$ and $\widehat{\mathbf{f}}(\mathbf{x}_j)$, where $i \geq j$. For the results of CIFAR-100 dataset on Figure 3, we show the histograms for the first 10 supervised classes due to the page width.

To understand more details of the feature representations, Figure 4 shows L2 norms of *unnormalized* feature representations extracted from the training data of CIFAR-10 and CIFAR-100. We used the same feature representation $\widehat{\mathbf{f}}(\mathbf{x})$ without L2 normalization as in Figures 2 and 3. Unlike cosine similarity and the collision upper bound values, the norm values increase by increasing $K$ on CIFAR-10 with all $K$ and CIFAR-100 with smaller $K$. We used the square root of the number of training samples, 223, as the number of bins of each histogram.

To show the tendency of cosine similarity and norms by increasing $K$, Figure 5 shows the relative change of a distance between histograms. Figure 5a shows the relative change values for the CIFAR-10 dataset based on Figures 2 and 4a. Similarity, Figure 5b shows the relative change values for the CIFAR-100 dataset based on Figures 3 and 4b. For the representations extracted from the CIFAR-10 dataset, we calculated the first Wasserstein distance between the histogram of $K + 1 = 32$ and the

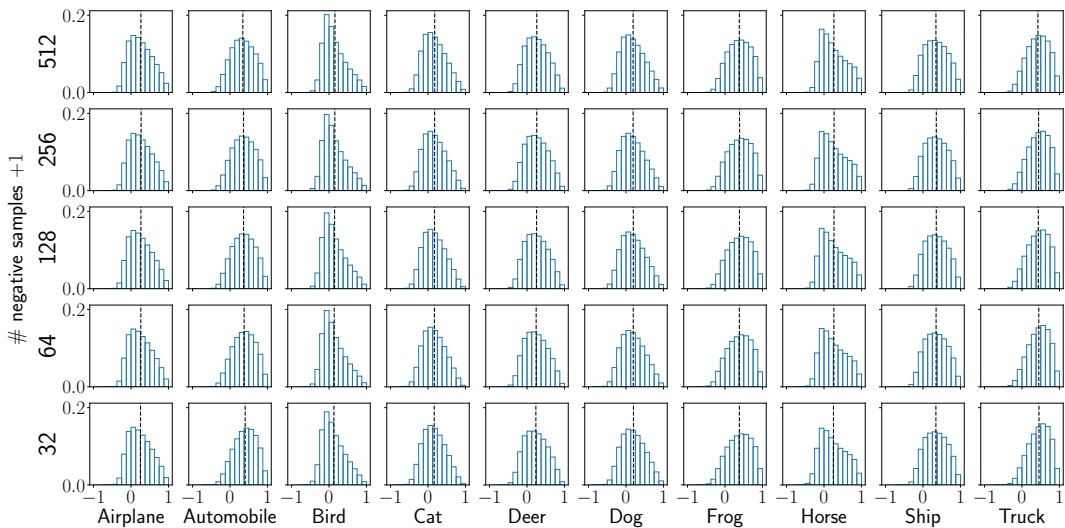

Figure 2: Histgrams of cosine similarity between the features in the same supervised class on CIFAR-10. The vertical lines represent the mean values.

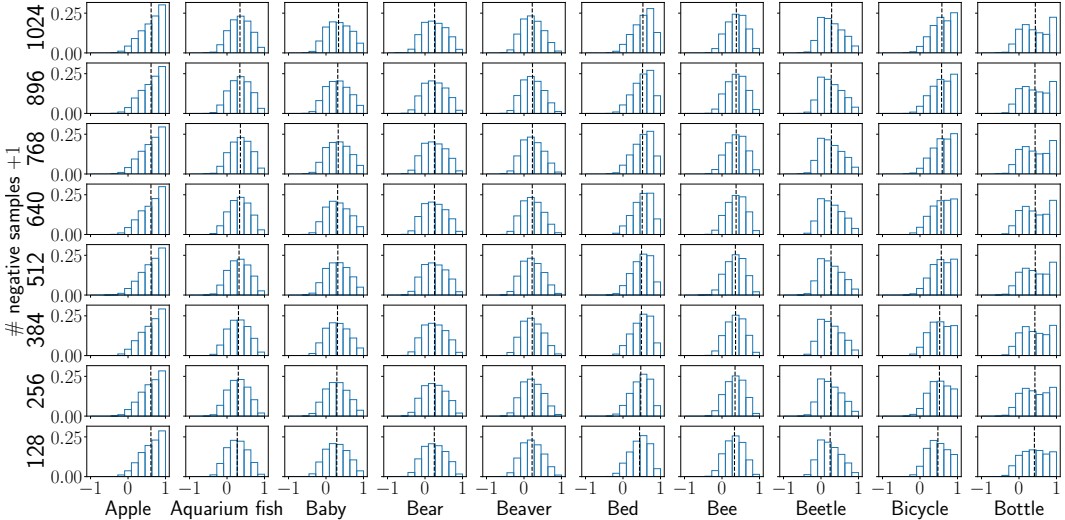

Figure 3: Histgrams of cosine similarity between the features in the same supervised class on CIFAR-100. The vertical lines represent the mean values.

other $K$ values with Scipy [Virtanen et al., 2020]. We used the distance between the histograms of $K + 1 = 32$ and $K + 1 = 64$ as the reference value of the relative change. We calculated the averaged Wasserstein distance values among the supervised classes by random seed and took the averaged values over the three different random seeds. For CIFAR-100, we used the Wasserstein distance between histograms of $K + 1 = 128$ and $K + 1 = 256$ as the reference value. Note that we used minimum and maximum norm values to unify the range of histograms among different $K$ since L2 norm values are not bounded above.

### F.2 Details of Collision bound Calculation on Table 1

To calculate the upper bound (11) of the collision term without coefficient terms $\alpha, \beta$, we sampled two data augmentations $\mathbf{a}$ and $\mathbf{a}^+$ per each training sample $\mathbf{x}$. The data augmentation distribution

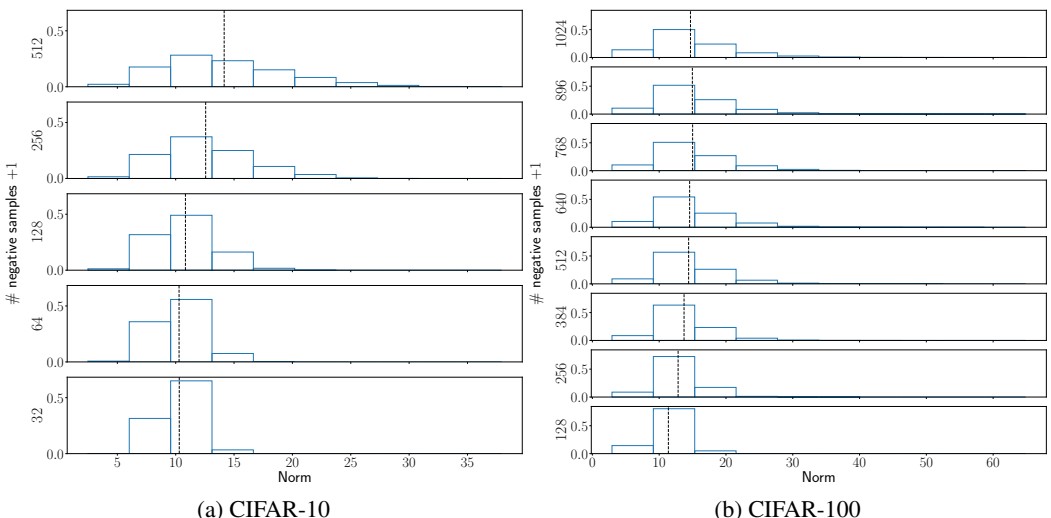

(a) CIFAR-10              (b) CIFAR-100

Figure 4: Histgrams of L2 norm values of unnormalized feature representations. The vertical lines represent the mean values.

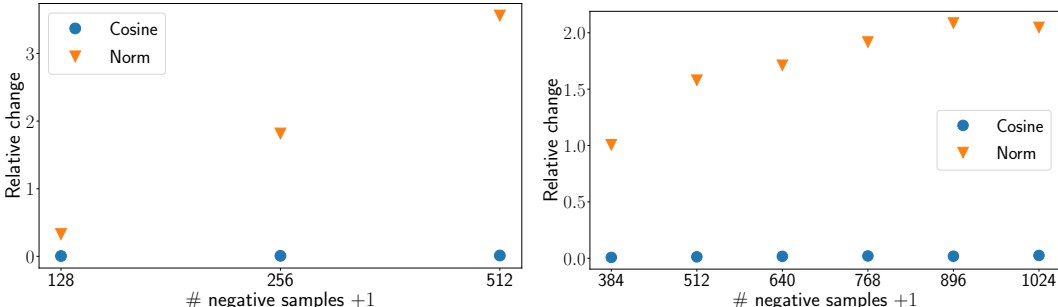

(a) Relative change of the first Wasserstein distance between the histogram of $K + 1 = 32$ and the histogram of the other $K + 1$ on CIFAR-10. The reference value is the distance between the histograms of $K + 1 = 32$ and $K + 1 = 64$.

(b) Relative change of the first Wasserstein distance between the histogram of $K + 1 = 128$ and the histogram of the other $K + 1$ on CIFAR-100. The reference is the distance between the histograms of $K + 1 = 128$ and $K + 1 = 256$.

Figure 5: Relative change of the first Wasserstein distance between histograms by increasing the number of negative samples.

was the same as described in Section 5. We approximated the upper bound term as follows:

$$\frac{1}{N} \sum_{i=1}^{N} \frac{1}{N_{y_i}} \sum_{j \neq i}^{N} \mathbb{I}\left[y_i = y_j\right] \times \left|\mathbf{f}(\mathbf{a}_i(\mathbf{x}_i)) \cdot \left[\mathbf{f}(\mathbf{a}_j(\mathbf{x}_j)) - \mathbf{f}(\mathbf{a}_i^+(\mathbf{x}_i))\right]\right|, \tag{27}$$

where recall that $N_y$ is the number of samples whose label is $y$. We reported the averaged value of Eq. (27) with respect to the random seeds on Table 1.

### F.3 Comprehensive Results of Table 1

Tables 4 and 5 show the comprehensive results of Table 1 for CIFAR-10 and CIFAR-100, respectively. Figure 6a shows upper bounds of supervised loss and the linear accuracy on the validation dataset.

### F.4 Experiments on Natural Language Processing

Arora et al. [2019] conducted experiments for contrastive unsupervised sentence representation learning on Wiki-3029 dataset that contains 3 029 classes. However, we cannot use this dataset to perform similar experiments to CIFAR-10/100. This is because we need a huge number of negative

Table 4: The bound values on CIFAR-10 experiments with different $K+1$. CURL bound and its quantities are shown with †. The proposed ones are shown without †. Since the proposed collision values are half of $^\dagger$Collision, they are omitted. The reported values contain their coefficient except for Collision bound.

| $K+1$ | | 32 | 64 | 128 | 256 | 512 |
|---|---|---|---|---|---|---|
| $\tau$ | | 0.96 | 1.00 | 1.00 | 1.00 | 1.00 |
| $\upsilon$ | | 0.69 | 0.99 | 1.00 | 1.00 | 1.00 |
| $\mu$ acc | | 72.75 | 75.30 | 77.22 | 78.60 | 80.12 |
| Linear acc | | 77.13 | 79.70 | 81.33 | 82.85 | 84.13 |
| Linear acc w/o | | 82.02 | 83.88 | 85.43 | 86.68 | 87.66 |
| $L_{\mathrm{Info}}$ | Eq. (2) | 2.02 | 2.64 | 3.29 | 3.96 | 4.64 |
| $d(\mathbf{f})$ | Eq. (7) | $-1.16$ | $-1.17$ | $-1.18$ | $-1.18$ | $-1.19$ |
| $^\dagger L_{\mathrm{Info}}$ bound | Eq. (8) | 0.23 | 0.76 | 1.41 | 2.08 | 2.75 |
| $^\dagger$Collision | | 1.32 | 1.93 | 2.58 | 3.26 | 3.94 |
| $^\dagger L_{\mathrm{sup}}^{\mu}$ | | 0.05 | 0.00 | 0.00 | 0.00 | 0.00 |
| $^\dagger L_{\mathrm{sub}}^{\mu}$ | | 0.01 | 0.00 | 0.00 | 0.00 | 0.00 |
| $L_{\mathrm{Info}}$ bound | Eq. (10) | 0.39 | 0.70 | 1.02 | 1.35 | 1.69 |
| $L_{\mathrm{sup}}^{\mu}$ | | 0.63 | 0.90 | 0.91 | 0.90 | 0.90 |
| $L_{\mathrm{sub}}^{\mu}$ | | 0.26 | 0.01 | 0.00 | 0.00 | 0.00 |
| Collision bound | Eq. (11) | 0.60 | 0.61 | 0.61 | 0.62 | 0.62 |

Table 5: The bound values on CIFAR-100 experiments with different $K+1$. CURL bound and its quantities are shown with †. The proposed ones are shown without †. Since the proposed collision values are half of $^\dagger$Collision, they are omitted. The reported values contain their coefficient except for Collision bound.

| $K+1$ | | 128 | 256 | 384 | 512 | 640 | 768 | 896 | 1024 |
|---|---|---|---|---|---|---|---|---|---|
| $\tau$ | | 0.72 | 0.92 | 0.98 | 0.99 | 1.00 | 1.00 | 1.00 | 1.00 |
| $\upsilon$ | | 0.00 | 0.00 | 0.15 | 0.62 | 0.90 | 0.98 | 1.00 | 1.00 |
| $\mu$ acc | | 32.74 | 34.22 | 35.27 | 35.98 | 36.58 | 37.10 | 36.84 | 37.50 |
| Linear acc | | 42.01 | 43.59 | 44.28 | 45.09 | 45.72 | 46.06 | 45.50 | 46.52 |
| Linear acc w/o | | 57.92 | 58.91 | 59.51 | 59.30 | 59.35 | 59.62 | 59.11 | 59.46 |
| $L_{\mathrm{Info}}$ | Eq. (2) | 3.32 | 3.98 | 4.38 | 4.66 | 4.88 | 5.06 | 5.21 | 5.34 |
| $d(\mathbf{f})$ | Eq. (7) | $-0.99$ | $-0.98$ | $-0.97$ | $-0.97$ | $-0.96$ | $-0.95$ | $-0.96$ | $-0.95$ |
| $^\dagger L_{\mathrm{Info}}$ bound | Eq. (8) | 0.72 | 0.47 | 0.58 | 0.79 | 0.98 | 1.15 | 1.29 | 1.43 |
| $^\dagger$Collision | | 0.69 | 1.15 | 1.48 | 1.73 | 1.93 | 2.10 | 2.24 | 2.37 |
| $^\dagger L_{\mathrm{sup}}^{\mu}$ | | 0.00 | 0.00 | 0.01 | 0.01 | 0.01 | 0.00 | 0.00 | 0.00 |
| $^\dagger L_{\mathrm{sub}}^{\mu}$ | | 1.02 | 0.30 | 0.07 | 0.01 | 0.00 | 0.00 | 0.00 | 0.00 |
| $L_{\mathrm{Info}}$ bound | Eq. (10) | 1.18 | 1.53 | 1.72 | 1.86 | 1.96 | 2.05 | 2.12 | 2.19 |
| $L_{\mathrm{sup}}^{\mu}$ | | 0.00 | 0.00 | 0.30 | 1.21 | 1.76 | 1.92 | 1.95 | 1.95 |
| $L_{\mathrm{sub}}^{\mu}$ | | 1.82 | 1.93 | 1.66 | 0.75 | 0.20 | 0.04 | 0.01 | 0.00 |
| Collision bound | Eq. (11) | 0.52 | 0.52 | 0.52 | 0.51 | 0.51 | 0.51 | 0.51 | 0.51 |

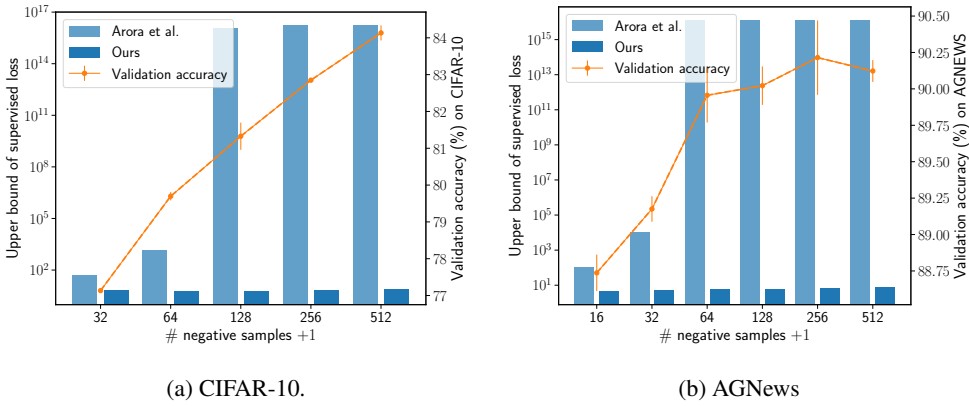

(a) CIFAR-10.  (b) AGNews

Figure 6: Upper bounds of supervised loss and validation accuracy.

samples to perform the same control experiments as CIFAR-10/100's experiments from the coupon collector's problem. Concretely, we need to more than $26\,030$ negative samples from Eq. (26). Such large negative samples cause an optimization issue in practice. In addition, unlike self-supervised learning on vision, self-supervised learning algorithms on text do not use a large number of negative samples in practice. For example, Logeswaran and Lee [2018], Gao et al. [2021] use at most $399$ and $1\,023$ negative samples in their experiments respectively.[13]

**Dataset and Data Augmentation**    We used the AGNews classification dataset [Zhang et al., 2015] due to the aforementioned difficulty of the experiment on the Wiki-3029 dataset. The dataset contains $4$ supervised classes and $120\,000$ training samples and $7\,600$ validation samples. As pre-processing, we used a tokenizer of torchtext with its default option: `basic_english`. After that, we removed words whose frequency was less than $5$ in the training dataset. As a data augmentation, we randomly delete $20\%$ words in each samples [Gao et al., 2021]. We tried a different data augmentation that randomly replaced $20\%$ words with one of the predefined similar words inspired by Wang and Yang [2015]. To obtain similar words, we used the five most similar words in pre-trained word vectors on the Common Crawl dataset [Mikolov et al., 2018]. If a word in the training data of the AGNews dataset did not exist in the pre-trained word vector's dictionary, we did not replace the word.

**Self-supervised Learning**    To compare the performance of supervised classification to the reported results on the AGNews dataset, we modify the supervised fastText model [Joulin et al., 2017] to model a feature encoder $\mathbf{f}$. The self-supervised model consists of a word embedding layer, an average pooling layer over the words, and the same nonlinear projection head as the CIFAR 10/100 experiment. The number of hidden units in the embedding layer and projection head was $50$.

We only describe the difference from the vision experiment because we mainly follow the vision experiment. We trained the encoder by using PyTorch on a single GPU because the training was fast enough on a single GPU due to the model's simplicity. The number of epochs was $100$. We used linear warmup at each step during the first $10$ epochs. We did not apply weight decay by following Joulin et al. [2017]. For the base learning rate $\texttt{lr} \in \{1.0, 0.1\}$ and initialization of learning rate, either $\texttt{lr} \times \frac{K+1}{256}$ or $\texttt{lr} \times \sqrt{K+1}$, which are used in Chen et al. [2020a].

**Linear Evaluation**    We followed the same optimization procedure as in the CIFAR 10/100 experiments except for the number of epochs that was $10$ and performing single GPU training. We used the mean classifier's validation accuracy as a hyper-parameter selection criterion since we perform grid-search among two types of data augmentation, two learning rates, and two learning rate initialization methods. Note that the deletion data augmentation performed better than the replacement one.

**Bound Evaluation**    Same as in the CIFAR 10/100 experiments except for $K + 1 \in \{32, 64, 128, 256, 512\}$ since the number of classes is $4$.

---

[13]Precisely, all other samples in the same mini-batch are treated as negative samples.

Table 6: The bound values on AGNews experiments with different $K + 1$. CURL bound and its quantities are shown with †. The proposed ones are shown without †. Since the proposed collision values are half of †Collision, they are omitted. The reported values contain their coefficient except for Collision bound.

| $K + 1$ | | 16 | 32 | 64 | 128 | 256 | 512 |
|---|---|---|---|---|---|---|---|
| $\tau$ | | 0.99 | 1.00 | 1.00 | 1.00 | 1.00 | 1.00 |
| $\upsilon$ | | 0.96 | 1.00 | 1.00 | 1.00 | 1.00 | 1.00 |
| $\mu$ acc | | 87.09 | 88.41 | 89.12 | 89.38 | 89.47 | 89.54 |
| Linear acc | | 88.74 | 89.18 | 89.96 | 90.02 | 90.21 | 90.12 |
| Linear acc w/o | | 87.11 | 87.94 | 89.05 | 89.49 | 89.62 | 89.45 |
| $L_{\text{Info}}$ | Eq. (2) | 1.23 | 1.77 | 2.37 | 3.02 | 3.69 | 4.37 |
| $d(\mathbf{f})$ | Eq. (7) | $-1.72$ | $-1.77$ | $-1.81$ | $-1.82$ | $-1.83$ | $-1.84$ |
| $^{\dagger}L_{\text{Info}}$ bound | Eq. (8) | $-0.22$ | 0.35 | 0.99 | 1.65 | 2.34 | 3.02 |
| $^{\dagger}$Collision | | 1.49 | 2.13 | 2.80 | 3.48 | 4.16 | 4.85 |
| $^{\dagger}L_{\text{sup}}^{\mu}$ | | 0.02 | 0.00 | 0.00 | 0.00 | 0.00 | 0.00 |
| $^{\dagger}L_{\text{sub}}^{\mu}$ | | 0.00 | 0.00 | 0.00 | 0.00 | 0.00 | 0.00 |
| $L_{\text{Info}}$ bound | Eq. (10) | $-0.39$ | $-0.10$ | 0.22 | 0.55 | 0.89 | 1.23 |
| $L_{\text{sup}}^{\mu}$ | | 0.57 | 0.61 | 0.63 | 0.64 | 0.64 | 0.65 |
| $L_{\text{sub}}^{\mu}$ | | 0.02 | 0.00 | 0.00 | 0.00 | 0.00 | 0.00 |
| Collision bound | Eq. (11) | 0.87 | 0.89 | 0.91 | 0.92 | 0.93 | 0.93 |
| $^{\dagger}$ $\ln L_{\text{sup}}^{\mu}$ upper bound | | 4.72 | 9.28 | 37.06 | 37.05 | 37.04 | 37.04 |
| $\ln L_{\text{sup}}^{\mu}$ upper bound | | 1.52 | 1.60 | 1.72 | 1.83 | 1.93 | 2.02 |

**Results**   Table 6 shows the quantities of bound-related values. When $K$ was too small, both lower bounds were vacuous because an InfoNCE value should be non-negative. However, the lower bounds were negative. This vacuousness comes from $d(\mathbf{f})$ that takes negative value. Figure 6b shows the upper bound of mean classifier and linear accuracy on the validation dataset by rearranging InfoNCE's lower bounds. Figure 6b shows the similar tendency to Figure 1; by increasing $K$, the existing bound explodes, but the proposed bound does not.