# OpenReview forum: "Understanding Negative Samples in Instance Discriminative Self-supervised Representation Learning"
_NeurIPS.cc/2021/Conference — NeurIPS 2021 Poster_

### Official Review · Reviewer_TgcC · 2021-07-16

**Rating:** 5
**Confidence:** 3

**Summary:**

The authors present some theoretical analysis that extend the analysis in CURL [Arora et al 2019]. The task they study is two steps: first learning an encoder with self-supervised learning (SSL) without labels and then freezing the encoder and learning linear classifiers with supervised learning (Sup) and using the labels on the same dataset. The analysis is trying to explain why having a large number of negative samples during contrastive self supervised pretraining with SimCLR  does not degrade the performance of the supervised second stage. They study the relationship between the two losses and validate their bounds on CIFAR.

**Limitations And Societal Impact:**

The authors do discuss some limitations, but not potential societal impacts. Given the nature of the work, the latter is not easy to assess and in my opinion it is fine to skip for a theoretical paper on SSL.

**Main Review:**

**Strengths**:
- The paper offers an interesting analysis on why performance does not decrease with more negatives for infoNCE, for a task that is  common in SSL papers, eg learning on imagenet without labels and then testing on imagenet after learning linear classifiers on the same training data with the labels.
- Overall the paper is well written and clear. Note that I did not go through the proofs in the appendix, but the main paper analysis seems reasonable.
 - The observations from Theorem 8 and the analysis on contributions from the sub-class loss is interesting.


**Weaknesses** (and questions):

A) The biggest weakness I think is that the analysis happens on a very restricted scenario, with no transfer: the authors study only the case where we have a single dataset and learn the encoder without using the label that we know exist and use to learn the classifiers - this is suboptimal and would not make sense in practive. I understand that this evaluation is common practice in SSL paper, however this is only a small part of the evaulations these papers have, and transfer learning is the more important and realisting setting. The authors do discuss this in lines 121-124 but justifying their choice by only citing empirical evidence of correlation of this "task" with transfer tasks, but I wouldn't say there are no guarantees there. Calling the second stage of classifier learning on the same dataset as traing as a  "downstream supervised task" is an exaggeration (I would suggest to the authors to rephrase). Although this task "correlates" with transfer tasks, it is not clear to me if also this analysis extends. It would be great to discuss this at least a bit further.

B) Even for this task above, there are further simplifications to facilitate the analysis: 1) only the SimCLR case is covered and yet, there is no analysis on a seemingly important (see SimCLR-v2 and other recent papers that show that) part of that approach, ie the projection head. 2) The MoCo approach which is a very popular variant with a memory queue is not discussed. _How does the analysis extend to negatives from a memory queue and dual encoders with exponential moving average?_  3) There is a further sumplification by the use of a mean classifier, which is not common practice . _Why is that simplification there, and is it central for the analysis?_

C) The (absolute) numbers in Table 1 are not so intutive, unbounded and hard to understand. It is really hard to understand what is the main message of Table 1 and some of the rows, eg colisions, could perhaps be made more informative by turning them into probabilities. It is unclear what is meant in line 269 by "10 sampled data augmentation per sample" and unclear what reporting the Collision bound without the alpha and beta constants offer (section 4.2 is very unclear to me).

Some more notes/questions:
- The discussion on clustering based SSL methods and Sec 4.4  is very restricted to this unrealistic task, that becomes even more unrealistic for clustering based pretraining. It is uncler to me what it offers.
- A missing ref (Kalantidis et al "Hard Negative Mixing for Contrastive Learning" NeuriPS 2020) synthesizes hard negatives for contrastive SSL. Same as MoCo, it would be interesting to discuss how this analysis extends to synthetic negatives.

**Rating**

Although an interesting study, the paper has limitations (see "weaknesses" section above). I would say that the current version of the paper is marginally below the acceptance threshold, but I am looking forward to the authors addressing my concerns above in their rebuttal.

-----------
## Post-rebutal thoughts

The authors provided extensive responses to my questions, answering many in a satisfactory way. I still think however that a central concern listed in the original review stand: the fact that Arora et al study the same task that first learns without labels and the with labels on the same dataset (and only that task) doesn't mean that this is what should be the only task to study for "Understanding Negative Samples in Instance Discriminative Self-supervised Representation Learning".

In their response, the authors claim that

> The self-supervised learning setting of our analysis is practical because the setting is quite similar to a semi-supervised learning setting, where we can access massive unlabeled samples and a few labeled samples.

With all due respect, I wouldn't compare this to semi-supervised learning for one key reason: as the authors also say here, in semi-supervised learning you have few labeled examples, a key property of the task. So, I would totally understand this analysis if the proposed bound was evaluated in a semi-supervised setting. This is not the case here, ie more than few labeled examples per class are used for learning the classifiers in this case.

Similarly, wrt the answer on the usage of a mean classifier:

> a few-shot learning setting uses a mean classifier, namely, Prototypical Networks [9], which has been cited more than 2700 times, according to Google Scholar.

Again, in the same way, the use of a mean classifier is indeed justified for few-shot learning, but it is well known that in the case of datasets with many labels, a logistic regression classifiers is superior.

Overall, I do see some merit in this paper, yet I think the breadth of the analysis is not enough; I will keep my score to 5.

**Time Spent Reviewing:**

4

---

> ### Author Response · Authors · 2021-08-10
> **Response to Reviewer TgcC (1/2)**
>
> We thank the reviewer for recognizing our paper’s strengths and giving constructive comments. We would like to address the comments and questions below.
>
> ## C1. Theoretical analysis is done under the restricted scenario: no transfer
>
> Our contribution is to fill the gap between the theoretical analysis by Arora et al. [2] and empirical observations of self-supervised representation learning such as [1] as in L37–41. The self-supervised learning setting of our analysis is practical because the setting is quite similar to a semi-supervised learning setting, where we can access massive unlabeled samples and a few labeled samples. In fact, self-supervised learning is more flexible than semi-supervised learning because we can annotate samples during self-supervised learning on the unlabeled samples in parallel. After the self-supervised learning, the feature extractor gives much better classification accuracy for a simple classifier trained on a few labeled samples than the classifier without the feature extractor or the supervised model from scratch. See also Table 7 in SimCLR paper [1] for empirical comparison under this scenario.
>
> In addition, the linear evaluation protocol is a defect-standard in self-supervised representation learning on ImageNet pre-training as you mentioned. Indeed, we can see abbreviation studies by varying the number of negative samples, for example, state-of-the-art contrastive learning papers such as SimCLR, MoCo, and SwAV, under the same evaluation scenario: both self-supervised learning and supervised learning on the same dataset. We arguably relax the setting in Arora et al. who perform the analysis with a no transfer setting as well; we can use data augmentation and do not access explicitly positive samples from the latent classes in our experiments. Furthermore, We still do not clearly understand self-supervised learning with negative samples even under this linear evaluation on the same dataset yet.
>
> Of course, we agree that the link to the transfer learning scenario is an important future direction. Fortunately, we think that our analysis can be plugged into a theoretical analysis of transfer learning, where the target task’s loss on a different dataset is bounded above by the source task’s loss, such as linear classifier’s loss. This is because the mean classifier’s loss is an upper bound of the linear classifier’s loss as discussed in the answer why the mean classifier-based analysis is not unrealistic in `C2.3. The mean classifier is not realistic` later.
>
> ---
>
> ### C1.1. Rephrasing `downstream supervised task`
>
> We believe that the term is commonly used in related work such as Arora et al. To avoid confusion with the existing work for the readers, we would like to keep the term. Moreover, we explicitly define the supervised task in Sec. 3.1.2 to clarify our theoretical setting.
>
> ---
>
> ## C2. Comments related to simplification
>
> ### C2.1. Projection head
>
> As we mentioned in L335–336 of the limitation paragraph, we are aware of the important role of discarding the projection head after self-supervised learning to improve the performance of supervised tasks in practice. However, we think that the procedure does not directly depend on the number of negative samples that is our main concern.
>
> We can add a linear classification performance without the projection head to Table 1 as reference. See __(without head) linear acc__ row in the following table, where _$\mu$ acc_ and _Linear acc_ are the same as in Table 1 for reference. We can see the classification performance is correlated with the performance of the mean classifier and linear classifier with the projection head.
>
> ||CIFAR-10|||||CIFAR-100||||
> |-:|-:|--:|-:|-:|-|-:|-:|-:|-:|
> |$K+1$|$32$|$128$|$256$|$512$||$128$|$256$|$512$|$1\,024$|
> |$\mu$ acc|$72.75$|$77.22$|$78.60$|$80.12$||$32.67$|$34.25$|$35.90$|$37.44$|
> |Linear acc|$77.13$|$81.33$|$82.85$|$84.13$||$41.95$|$43.53$|$45.16$|$46.57$|
> |__(without head) linear acc__|$82.02$|$85.43$|$86.68$|$87.66$||$57.98$|$59.08$|$59.23$|$59.50$|
>
> ---
>
> ### Q2.2. Can we apply the proposed analysis to MoCo?
>
> Yes, we can. Our main theoretical results are valid with asymmetric feature extractors: the positive and negative features are extracted by using a different feature extractor such as another neural network or memory bank used in MoCo instead of the same feature encoder for $\mathbf{f}(\mathbf{a}(\mathbf{x}))$. This is because both positive and negative features are aggregated into $\boldsymbol{\mu}\_{c^+}$ and $\boldsymbol{\mu}\_{c^-}$ by using Jensen’s inequality in our analysis. We would like to mention this fact in our theorem.
>
> ---
>
> ### C2.3. The mean classifier is not realistic
>
> A more general classifier’s supervised loss is bounded above by the mean classifier’s supervised loss as described in L113–114. Thus we believe that the analysis of the mean classifier is well-motivated.
> From the empirical perspective, a few-shot learning setting uses a mean classifier, namely, `Prototypical Networks` [9], which has been cited more than 2700 times, according to Google Scholar. For example, Table 4 in [10] and `ProtoCLR` with `ProtoNet` testing rows on Tables 3–5 in [11] report Prototypical Networks-based classifier’s performance with self-supervised representation learning. Especially, in [11], the mean classifier’s performance is competitive to fine-tuning counterpart in practice. This few-shot learning setting is also similar to the practice setting as answered in C1.: we only require few labeled samples. Moreover, the mean classifier is feasible in a CPU environment because it does not require any updates of the weights.
>
> In terms of comparison to the existing work, Arora et al. also use the mean classifier as the center of their analysis. Moreover, recent work still analyzes the mean classifier’s, see Sec. 6 in [8].

---

> ### Author Response · Authors · 2021-08-10
> **Response to Reviewer TgcC (2/2)**
>
> ## C3. Clarification of Table 1.
>
> Table 1 gives the fine-grained numerical values used in Figure 1. The main messages from Table 1 (and Figure 1) are
>
> - By increasing the number of negative samples $K$, the supervised accuracy also increases, see related rows _$\mu$ acc_ and _Linear acc_.
> - However, the existing bound values, _$^\dagger L\_{\mathrm{Info}}$ bound_, are dominated by _$^\dagger$ Collision_ that is not related to the supervised loss by increasing $K$. Thus we cannot draw the conclusion that increasing $K$ gives better performance on supervised learning.
> - On the other hand, the proposed bound values, _$L\_{\mathrm{Info}}$ bound_, contain $L^{\mu}_{\mathrm{sup}}$ with large proportion by increasing $K$.
>
> We mentioned this bound comparison in L281-L283.
>
> ---
>
> ### Q3.1. Can Collision row be replaced with probability?
>
> No. The _$^\dagger$Collision_ row represents $\tau_K \mathbb{E} [\ln (\mathrm{Col} + 1) \mid \mathrm{Col} \neq 0 ]$ in Eq. 8; this is not a probability value. But we can add the both probabilities $\tau_K$ and $\upsilon_{K+1}$ to Table 1 as follows:
>
> |                           |  CIFAR-10  |          |        |       |   | CIFAR-100 |        |         |          |
> |--------------------------:|-----------:|---------:|-------:|------:|---|----------:|-------:|--------:|---------:|
> |           $K+1$           |  $32$      | $128$    |  $256$ | $512$ |   |     $128$ |  $256$ |   $512$ | $1\,024$ |
> | $\tau$                    |  $0.96$    |   $1.00$ | $1.00$ |$1.00$ |   |    $0.72$ | $0.92$ |  $0.99$ |   $1.00$ |
> | $\upsilon$                |  $0.69$    |   $1.00$ | $1.00$ |$1.00$ |   |    $0.00$ | $0.00$ |  $0.60$ |   $1.00$ |
>
> Note that the following difference between $\tau$ and $\upsilon$:
>
> - Higher $\tau$ is worse since the existing lower bound is dominant by _collision term_ in Eq. 8 that is not related to supervised loss at all.
> - Higher $\upsilon$ is better since the proposed lower bound is dominant by _supervised loss_ in Eq. 10.
>
> ---
>
> ### Q3.2. What is meant in line 269 by "10 sampled data augmentation per sample"?
>
> To evaluate $\boldsymbol{\mu}_{c} = \mathbb{E}\_{\mathbf{x} \sim \mathcal{D}\_{c}} \mathbb{E}\_{\mathbf{a} \sim \mathcal{A}}  \mathbf{f} (\mathbf{a}(\mathbf{x}))$ for mean classifier’s loss, we need sampling approximation over the data augmentation since we cannot evaluate the expectation in $\boldsymbol{\mu}\_{c}$ explicitly. Thus we sampled 10 data augmentations, $\\{ \mathbf{a}\_j \\}\_{j=1}^{10} \sim \mathcal{A}^{10}$, for each training sample $\mathbf{x}$ and then we averaged them over the same latent class $c$. More precisely, for each $c$, $\frac{1}{N\_c} \sum\_{i=1}^N \mathbb{I} [y\_i = c] \frac{1}{10} \sum\_{j=1}^{10} \mathbf{f}( \mathbf{a}\_{j}(\mathbf{x}\_i))$, where $N\_c$ is the number of samples whose latent class is $c$, and other notations are same as in our notation table (Table 2). If the equation makes the understanding clearer, we would like to add this equation to the L269 as a footnote to the manuscript.
>
> ---
>
> ### Q3.3. What reporting the Collision bound without the $\alpha$ and $\beta$ constants offer and clear explanation of Sec. 4.2?
>
> The main concern in the Sec. 4.2 is what happens on the feature representations when $K$ increases by looking at the upper bound of the collision term. Intuitively, we expect that a large number of negative samples enlarge dissimilarity between feature representations in the same latent class since the collision term becomes larger as shown in Eq. 11 and Lemma 4.4 by Arora et al.
> However, as in the row of _Collision bound_ in Table 1, such enlarging dissimilarity does not happen in practice due to the formulation of InfoNCE loss as discussed in L194-201 and Appendix D.
>
> ---
>
> ## C4. Unclear points
>
> ### C4.1 The discussion on clustering based self-supervised learning methods and Sec 4.4 is restricted and unclear
>
> As we answered in C1. and C2.3., the ImageNet-based evaluation protocol is a de-facto standard evaluation procedure in self-supervised representation learning research. It is a realistic setting. Could you explain a little bit more about which part is unrealistic for clustering based pre-training?
>
> By performing the clustering, clustering-based algorithms replace InfoNCE loss over instance-wise classification ($K+1$ classification) with pseudo-label classification obtained by performing clustering over the feature representations; they decouple the number of negative samples and the mini-batch size. As a result, if the obtained clusters approximate the supervised classes well, the InfoNCE loss can be a good approximation of supervised loss with a smaller mini-batch size, unlike SimCLR. On the other hand, SimCLR requires a large mini-batch size that is directly related to the number of negative samples. Our analysis offers that this is why the clustering-based self-supervised learning algorithm, e.g., SwAV, can perform better than SimCLR with a smaller mini-batch size as reported in the SwAV paper.
>
> ---
>
> ### C4.2 Discuss how the analysis extends to synthetic negatives [12]
>
> We think that the effect of collision term can decrease by creating hard negative features can be mixed feature representations whose latent classes differ in MoCHi [12], for example, $c^{-}_1 = c^+$, but $c^{-}_2 \neq c^+$. Thus we would like to add this paper [12] and related work to Sec. 6 in our manuscript.
>
> We note that including the given reference [12], _hard negative mining_ in metric learning and contrastive learning papers discuss the __quality__ of negative sampling to make the training more effective to avoid using false negative or too easy negative samples. On the other hand, our concern is the __quantity__ of negative samples to fill the gap between the existing theoretical analysis and empirical observation in self-supervised learning. We agree that the combination of both approaches is a fruitful direction to improve contrastive learning from both theoretical and empirical perspectives.
>
> ---
>
> [8] Chuang et al. Debiased Contrastive Learning. In _NeurIPS_, 2020.
>
> [9] Snell et al. Prototypical Networks for Few-shot Learning. In _NeurIPS_, 2017.
>
> [10] Su et al. When Does Self-supervision Improve Few-shot Learning?. In _ECCV_, 2020.
>
> [11] Medina et al. Self-Supervised Prototypical Transfer Learning for Few-Shot Classification. _arXiv_, 2020.
>
> [12] Kalantidis et al. Hard Negative Mixing for Contrastive Learning. In _NeurIPS_, 2020.

---

> ### Author Response · Authors · 2021-09-01
> **Re: Post-rebuttal thoughts**
>
> Dear Reviewer TgcC,
>
> We appreciate that the reviewer is satisfied with our response and shares the remained concerns. We would like to clarify the remained ones.
>
> ### Theoretical scenario
> We do not assume that all training samples are labeled to learn classifiers and to evaluate the bound. This is because we can approximate the mean classifier’s weight $\boldsymbol{\mu}_c$ with a few labeled samples per latent class instead of using all representations from the training dataset. Indeed, we can report the bound values when we use a few labeled samples per latent class to construct a mean classifier whose feature extractor is learned on the _unlabeled_ training dataset. Similar experiments are shown by the $\mu$–$5$ column of Table 1 in Arora et al. More empirically, Table 7 in SimCLR paper reports such a few labeled experimental results with linear classifier after self-supervised representation learning on the unlabeled dataset. Thus our setting is arguably realistic and similar to the existing weakly supervised learning frameworks such as semi-supervised learning or few-shot learning. We are happy to add such a table for our manuscript.
>
> ---
>
> ### Mean classifier
>
> In our first response, we should have emphasized that the linear classifier’s loss is bounded above by the mean classification’s loss as in L113-114 of our manuscript. Moreover, the mean classifier is a special case of linear classifier as discussed in Sec. 2.4 of [9]. Note that this property is not limited to few-shot learning alone. Thus we can link a linear classifier’s quantity with the mean classifier’s quantity in our results seamlessly. Concretely, our main theorem, Theorem 8, with linear classifier $ \mathbf{g}$ is by using $L\_{\mathrm{sup}}(\widehat{\mathbf{g}} \circ \mathbf{f}) \leq L^{\mu}\_{\mathrm{sup}}(\mathbf{f})$, where $\widehat{\mathbf{g}} = \mathrm{argmin}\_{\mathbf{g} }  L\_{\mathrm{sup}}(\mathbf{g} \circ \mathbf{f})$ as follows :
>
>
> $L_{\mathrm{Info}}(\mathbf{f})$
>
> $\geq \text{the right hand side of Eq. 10}$
>
> $\geq \frac{1}{2} \Big\\{$
> $\upsilon\_{K+1}  \mathbb{E}\_{c, \\{c^-\_k\\}_{k=1}^K \sim \rho^{K+1} }$
> $ [ L\_{\mathrm{sub}}(\widehat{\mathbf{g}} \circ \mathbf{f}, \mathcal{C})
>     \mid \mathcal{C}\_{\mathrm{sub}} = \mathcal{C} ]$
>
> $\hspace{2mm} + (1-\upsilon\_{K+1}) \mathbb{E}\_{c, \\{c^-\_k\\}\_{k=1}^K \sim \rho^{K+1} }$
> $[L\_{\mathrm{sub}}(\widehat{\mathbf{g}} \circ \mathbf{f}, \mathcal{C}\_{\mathrm{sub}} )
>     \mid \mathcal{C}\_{\mathrm{sub}} \neq \mathcal{C}
>     ]$
>
> $\hspace{2mm} + \mathbb{E}\_{c, \\{c^-\_k\\}\_{k=1}^K \sim \rho^{K+1}}
>     \ln (\mathrm{Col} + 1)$
> $ \Big\\} + d( \mathbf{f} ).$
>
>
> Related to our clarification of the first point above, we can find similar points between our analysis including Arora et al.’s one and few-shot learning’s empirical setting: the usage of mean classifier and data assumption. However, we focus on the popular empirical evaluation by varying the number of negative samples in self-supervised learning: linear evaluation on the same dataset. To highlight our contribution to filling the gap between the theoretical analysis by Arora et al. and the recent empirical results, the rigorous few-shot and semi-supervised learning settings are out-of-scope. Of course, we are happy to add this future direction to Conclusion section.

---

> ### Author Response · Authors · 2021-09-09
> **We are wondering if you are satisfied with our additional response**
>
> Dear Reviewer TgcC,
>
> Since we had posted the clarification on the remaining concerns on 1st Sept, we could not have seen an additional update. We hope that you are satisfied with our clarification. As additional information, please see also the response to the Reviewer QhKs, https://openreview.net/forum?id=pZ5X_svdPQ&noteId=_CH9vggHQ1K, for additional difficulties to analyze on transfer learning settings.
>
> We are wondering if you would mind updating the review again.

---

### Official Review · Reviewer_QhKs · 2021-07-18

**Rating:** 6
**Confidence:** 1

**Summary:**

This paper provides a new theoretical framework to analyze an empirical observation common in self-supervised representation learning, i.e., using a large number of negative samples than the number of supervised classes can improve performance. However, previous work for contrastive unsupervised representation learning cannot explain above observation theoretically. Therefore, the paper proposes a new lower bound using the coupon collector’s problem, and confirms this analysis on the CIFAR-10/100 datasets.

**Limitations And Societal Impact:**

Please see my main review above.

**Main Review:**

Previous works on theoretical framework of contrastive unsupervised representation learning cannot explain the empirical observation that increasing the number of negative samples can improve performance. This paper proposes to analyze it with a novel lower bound with the help from the coupon collector’s problem of probability theory. The methodology part is organized in a clear form. Writing is good and easy to follow. The proposed bound is also evaluated in two CV datasets.

However, the reviewer still has following three concerns.
First, comparing to the important related work, i.e., CRUL (Arora et al., 2019), this paper only conducts experiments on image domains but ignoring text domains. More experiments on other domains should be investigated for robust performance and validation.

Second, Figure 1 is still unclear. It is mentioned in the second page, while readers have to read all equations until eq.(8) and (10) in fifth page to understand the meaning. Besides, the legend is confusing, why not using different color for left/right bars.

Third, recent work of BYOL (Grill et al., 2020) and SimSaim (Chen et al., 2021) build self-supervised learning framework without negative samples, which are not discussed in this paper.

### Post-rebuttal and discussions

Thanks to authors' responses. Actually, their response has answered most of my concerns.

Besides my previous concerns, I also agree with other reviewers that the proposed theoretical framework is limited to no-transfer tasks in self-supervised learning.
Although in authors’ response, they pointed out the proposed framework can be plugged into transfer learning case. Still, it requires a thorough analysis and empirical evaluation.
I suggest authors to discuss above limitations in the paper, since they use a rather ambitious title.

However, as mentioned in authors' response, this paper indeed provides a theoretical explanation on some important observations (increasing number of negatives helps improving accuracy) in previous works (SimCLR, MoCo, etc.), which is untouched in CRUL. Therefore, I choose to increase the score to 6, marginally above the borderline.

I also think authors should try to demonstrate the main take-away in the introduction part, instead of the current statement in L40-41, which can possibly help readers better understanding their contributions.


**Time Spent Reviewing:**

6

---

> ### Author Response · Authors · 2021-08-10
> **Response to Reviewer QhKs**
>
> We thank the reviewer for suggestions and comments. We address the comments below.
>
> ## C1. Comparison on text domains like Arora et al. [2]
>
> As far as we know, a large negative samples setting is not common in self-supervised representation learning for natural language processing. Our theoretical analysis is motivated by the well-known empirical evidence from self-supervised representation learning on the vision domain as described in Sec. 3.3. In addition, in our understanding, the widely used self-supervised learning algorithms in natural language processing, such as BERT [3], RoBERTa [4], and GPT-3 [5], heavily depend on masked-language modeling tasks rather than solving instance discriminative tasks that are our target of the analysis. We also emphasize that the issue in the existing bound by Arora et al. does not depend on the domain; it apparently happens due to the definition of $\tau_K$ as in Sec. 3.3.
>
> From a more experimental view, we cannot use the same dataset, Wiki-3029 whose classes are $3\,029$, as in Arora et al. This is because we need a huge number of negative samples to perform the same control experiments as CIFAR-10/100’s experiments from the coupon collector’s problem. Concretely, we need to more than $26\,030$ negative samples from Eq. 25 in Supplementary Material. Such large negative samples cause an optimization issue in practice.
>
> ---
>
> ## C2. Figure 1 is still unclear: Why not use the different colors for left/right bars?
>
> We exactly follow our ICML2021 reviewer’s comment to revise Figure 1 as described in `Submission History - Improvements Made`. Of course, we appreciate your suggestion of improving the legend. We would like to remove the star marks and add both colors with bounds’ names to the legend.
>
> We believe that we explain the problem of the existing bound in the caption of Figure 1: L56–L63. We are wondering if you could give more concrete suggestions to improve Figure 1. We would like to revise Figure 1 by following your suggestions.
>
> ---
>
> ## C3. No discussion for BYOL and SimSiam algorithms that do not use any negative samples.
>
> We analyze the effect of the number of negative samples in instance-discriminative (or contrastive) representation learning, where negative samples must be necessary, as in our contributions in L37–41. Our analysis has the potential to give an insight into the related research area because contrastive learning is used beyond representation learning such as metric learning [6]. Both BYOL and SimSiam, which are cited as recent self-supervised learning algorithms in L15, do not use any negative samples. Thus clearly, both algorithms are out of the scope of our and Arora et al.’s analyses. We can mention that our analysis does not cover such algorithms in the Limitation paragraph of Sec. 7. For these algorithms, we can also mention the recent theoretical analysis [7] that appears after this NeurIPS submission as an accepted paper at ICML 2021.
>
>
> ---
>
> [3] Devlin et al. BERT: Pre-training of Deep Bidirectional Transformers for Language Understanding. In _NAACL-HLT_, 2019.
>
> [4] Liu et al. RoBERTa: A Robustly Optimized BERT Pretraining Approach. _arXiv_, 2019.
>
> [5] Brown et al. Language Models are Few-shot Learners. In _NeurIPS_, 2020.
>
> [6] Sohn. Improved Deep Metric Learning with Multi-class N-pair Loss Objective. In _NeurIPS_, 2016.
>
> [7] Tian et al. Understanding self-supervised Learning Dynamics without Contrastive Pairs. In _ICML_, 2021.

---

> > ### Author Response · Authors · 2021-08-27
> > **Could you discuss the concerns?**
> >
> > Dear Reviewer QhKs,
> >
> > If the reviewer still recognizes the raised points as concerns, we would like to clarify them since this NeurIPS discussion is the rolling style and we believe that we have already addressed all concerns.
> >
> > We provide additional experimental results to address one of the concerns, C1, from an empirical perspective as follows.
> >
> > ## Additional NLP experiments
> >
> > Even though there exists the difficulty of comparison to the Arora et al. on the text domain as in C1 of our first author feedback due to the several reasons: contrastive learning with large $K$ is not widely used approach, too large architecture, and inappropriate dataset, we could conduct an additional experiment on the text dataset, AG News that is a news topic classification dataset. In our experiments, linear classifiers with self-supervised representations achieve comparable validation accuracy to the fully supervised algorithms in Table 1 of [13]. If the experimental results make our contribution more robust, we are happy to add the results to the main body or Appendix of our manuscript.
> >
> > The following table shows the linear classifier’s validation accuracy and upper bounds of supervised loss on the text dataset.
> >
> > | $K+1$                         |                                |   $32$ |    $64$ |   $128$ |   $256$ |   $512$ |
> > |-----------------------------|--------------------------------|-------:|-------:|-------:|-------:|-------:|
> > | Linear acc ($\uparrow$)                 |                                | $89.18$ | $89.96$ | $90.02$ | $90.21$ | $90.12$ |
> > | $^{\dagger} \ln L^{\mu}_{\mathrm{sup}}$ upper bound ($\downarrow$) |                               |  $9.28$ | $37.06$ | $37.05$ | $37.04$ | $37.04$ |
> > | $\ln L^{\mu}_{\mathrm{sup}}$ upper bound ($\downarrow$)  |                                | $1.60$ |  $1.72$ |  $1.83$ |  $1.93$ |  $2.02$ |
> >
> > The tendency is similar to our CV experiments shown in Figure 1 when the number of negative samples $K$ increases. By increasing $K$, the validation accuracy of a linear classifier also improves. But the existing upper bound, $^{\dagger} \ln L^{\mu}\_\{\mathrm{sup}\}$, easily becomes large by increasing $K$. Clearly, this is because $\tau_K$ easily converges to $1$ by increasing $K$ from its definition. As a result, the collision term is dominant in the upper bound. Note that the bound values are the log scale.
> > On the other hand, the proposed upper bound, $\ln L^{\mu}_{\mathrm{sup}}$, does not still become like the existing upper bound. The experimental details are found in this reply post.
> >
> > ---
> >
> > ## Experimental Details
> >
> > We describe the experimental details by following `Sec. 5 Experiments` in our manuscript.
> >
> > ### Dataset and Data Augmentation
> >
> > We used the AG News classification dataset. The dataset contains $4$ supervised classes and $120\ 000$ training samples and $7\ 600$ validation samples. As pre-processing, we used a tokenizer of `torchtext` with its default option: `basic_english`. After that, we removed words whose frequency was less than $5$ in the training samples. As a data augmentation, we randomly delete $20\\%$ words in each sample [14]. We tried a different data augmentation that randomly replaced $20\\%$ words with their similar words inspired by [15]. To obtain similar words, we used the five most similar words in pre-trained word vectors on the Common Crawl dataset [16]. If a word in the training data of the AG News dataset did not exist in the pre-trained word vectors, we did not replace such a word.
> > In the preliminary experiments, the deletion data augmentation performed better than the replacement one.
> > Therefore we reported the deletion's results.
> >
> > ### Self-supervised Learning
> >
> > To compare the performance to the reported results on the AG News dataset, we modify the supervised fastText model [13] to model a feature encoder. The self-supervised model consists of a word embedding layer, average pooling layer over the words, and the same nonlinear projection head as in the CIFAR 10/100 experiment. The number of hidden units in the embedding layer and projection head was $50$.
> >
> > We only describe the difference from the vision experiment because we mainly follow the vision experiment. We trained the encoder by using PyTorch on a single GPU. The number of epochs was $100$. We used linear warmup at each step during the first $10$ epochs. We did not apply weight decay by following [13]. For the base learning rate in $\text{base lr} \in [1.0, 0.1]$ and initialization of learning rate, either $\text{base lr} \times (K+1) / 256$ or $\text{base lr} \times \sqrt{K+1}$, which is inspired by SimCLR paper.
> >
> > ### Linear Evaluation
> >
> > We followed the same optimization procedure as in the CIFAR 10/100 experiments except for the number of epochs that was $10$ and performing single GPU training. We used the mean classifier's validation accuracy as a hyper-parameter selection criterion since we perform grid-search on learning rate and its initialization methods.
> >
> > ### Bound Evaluation
> >
> > Same as in the CIFAR 10/100 experiments except for $K+1 \in \\{32, 64, 126, 256, 512 \\}$ since the number of classes is $4$.
> >
> >
> > ---
> > ### References
> >
> > [13] Joulin et al. Bag of Tricks for Efficient Text Classification. In _EACL_, 2017.
> >
> > [14] Gao et al. SimCSE: Simple Contrastive Learning of Sentence Embeddings. _arXiv_, 2021.
> >
> > [15] Wang & Yang. That’s So Annoying!!!: A Lexical and Frame-Semantic Embedding Based Data Augmentation Approach to Automatic Categorization of Annoying Behaviors using #petpeeve Tweets. In _EMNLP_, 2015.
> >
> > [16] Mikolov et al. Advances in Pre-Training Distributed Word Representations. In _LREC_, 2018.

---

> ### Author Response · Authors · 2021-09-08
> **Hope for rolling discussion**
>
> Dear Reviewer QhKs,
>
> As in Reviewer Guidelines, this NeurIPS discussion is a rolling style to minimize the chance of misunderstandings on the submitted papers by reviewers. So we, the authors, would like to know whether or not we have addressed all concerns raised by the reviewer in our first feedback and the additional feedback providing experimental results to address one of the raised concerns from an empirical perspective (https://openreview.net/forum?id=pZ5X_svdPQ&noteId=mj12B3feNyr). However, we could not have seen additional feedback from the reviewer for this NeurIPS discussion period; we hope that the reviewer was happy with our author's feedback.
>
>
> Thus we really appreciate that the reviewer could revise the rating. Otherwise, we would like to know the remaining concerns and clarify them.

---

> ### Author Response · Authors · 2021-09-09
> **Re: Post-rebuttal**
>
> Dear Reviewer QhKs,
>
> We appreciate that the reviewer is satisfied with our response and shares the remaining concern. We would like to clarify it.
>
> As in the comment by the reviewer TgcC and our feedback to the reviewer TgcC, our theoretical setting is common in both empirical and theoretical work. Please see the response to the reviewer TgcC for the empirical details.
>
> We certainly believe that we contribute to the existing problem regardless of transfer learning: filling the gap between the existing theoretical analysis by Arora et al. and the empirical observation when the number of negative samples increases as in L37–41. We agree that performing the analysis with transfer learning is an interesting direction. However, performing analysis with transfer learning makes our contributions vague and is not a promising way to improve our manuscript. This is because we still do not understand self-supervised learning well under our setting even though our setting seems simple. To move forward in the theoretical understanding of self-supervised learning, we need to discuss the existing issue first under this simple but practical setting. Indeed the existing work by Arora et al. also only focuses on the no transfer setting with stronger assumptions than ours. Notably, we relax assumptions than Arora et al. to make the theoretical setting more practical. Concretely, our analysis (1) allows us to use data augmentation and (2) does not require explicit positive pairs from the same latent class. Thanks to relaxing the assumptions, we can analyze recent self-supervised learning algorithms such as SimCLR.
>
> In addition, it is really difficult to add an analysis with transfer learning to our paper due to the page limitation of the conference paper because we need to formulate transfer learning with self-supervised learning and additional experiments as you described. Moreover, theoretical analysis of transfer learning is still actively proposed in the machine learning community. There is no dominant theoretical framework to understand transfer learning itself yet. Thus we would like to make our theoretical analysis as simple as possible for future work rather than combining one of the existing transfer learning frameworks with our analysis.

---

### Official Review · Reviewer_Rz1c · 2021-07-19

**Rating:** 6
**Confidence:** 3

**Summary:**

The authors point out a scenario where existing theory for contrastive self-supervised learning does not match up with what is seen empirically. Namely, that the work of Arora et al. (2019) predicts that increasing the number of negative samples will result is poorer generalisation performance, but the opposite is often seen in practice. The authors adapt the framework of Arora et al. (2019) to improve in this area, and experimentally show that their bound is tighter when the number of negative samples grows.

**Limitations And Societal Impact:**

There is some discussion of the limitations, where they mention that their approach does not take into account data augmentation explicitly. Looking at Table 1, I think another limitation is that their analysis does not completely solve the problem they identified with the work of Arora et al. (2019), though they do seem to have tighter bounds.

**Main Review:**

The main goal of the paper is to improve the theoretical analysis from Arora et al. (2019) to better match phenomena that are seen in practice. This is accomplished using an argument based on the coupon counter's problem. I think the main take-away of the paper is an interesting contribution, but I find the derivations somewhat hard to follow in some places so I'm not sure if my assessment is entirely accurate. I would appreciate if the authors could answer a few questions so I can check my understanding of the paper:
* What is the relationship between latent classes and supervised classes? From what I can tell, it is assumed that the set of supervised classes will be a subset of the latent classes, but would it not be more realistic to say that supervised classes are mixtures/unions of latent classes?
* Definition 7 requires $\rho$ to be a uniform distribution over latent classes, but from my understanding Arora et al. (2019) do not require this. What are the possible negative implications of this additional assumption?
* Table 1 and Figure 1 seem to be at odds: the table shows that the proposed bound increases (albeit slower) as a function negative samples, but Figure 1 shows a peak and then decrease.

For now, I have set my rating to marginally below the acceptance threshold because of the concerns outlined in the questions above. I am open to revising this score upwards after reading feedback from the authors.

Post response: my concerns have been somewhat addressed, so I have increased my score.

**Time Spent Reviewing:**

6

---

> ### Author Response · Authors · 2021-08-10
> **Response to Reviewer Rz1c**
>
> We thank the reviewer for recognizing our contribution to the theoretical analysis of self-supervised representation learning. To clarify the unclear points for the reviewer, we address the questions below.
>
> ## Q1. Do you assume that a supervised class is a subset of the latent classes? How about supervised classes that are mixtures/unions of latent classes?
>
> Yes, you are right. We assume that the supervised classes are subsets of latent classes to focus on the common evaluation procedure in self-supervised representation learning. We think that our assumption of class relationship is not weaker than Arora et al. [2].
>
> We believe that our main theorem (Theorem 8) covers the setting, where supervised classes are unions of latent classes by replacing $\mathcal{C}$ with $\mathcal{Y}$ that are unions of latent classes. It is worth noting that the classifier’s accuracy on all latent classes is lower or the same accuracy at worse as the same classifier on the supervised classes such that each supervised class is a union of the latent classes. For example, suppose the latent classes are breeds of dog and cat of _Oxford Pet_ image classification dataset and supervised classes are binary: dog and cat, where each binary label is the union of the latent classes. The classifier on supervised classes can mispredict the class labeled among dog breeds for a dog image, thus the fine-grained latent classes are beneficial for the classification on coarse-grained supervised classes: the union of the latent classes.
>
> ---
>
> ## Q2. What are the possible negative implications of Definition 7 requiring $\rho$ to be a uniform distribution over latent classes?
>
> There is no negative implication since Theorem 8 is valid without the assumption of $\rho$. Definition 7 gives the explicit form of the probability $\upsilon_K$ for the better understanding of $\upsilon_K$. Although this explicit $\upsilon_K$ is used in our experiment, where both $\rho$ for CIFAR-10/100 are uniform (please refer to L272–273), we would like to move Definition 7 to Appendix C in the future version if Definition 7 misleads the readers.
>
> ---
>
> ## C3. Table 1 and Figure 1 seem to be at odds
>
> Table 1 and Figure 1 are consistent. In Table 1, _$L_\mathrm{Info}$ bound_ row represents __lower bounds__ of the InfoNCE loss defined by Theorem 8. On the other hand, Figure 1 shows __upper bounds__ of $L^{\mu}_{\mathrm{sup}}$, precisely, the following value
>
> $\frac{1}{\upsilon_{K+1}} \left[ 2 L_{\mathrm{Info}}(\mathbf{f})  - (1-\upsilon_{K+1}) \\mathbb{E}\_\{ c, \\{ c\_k^- \\}^K\_\{k=1\} \} L^\mu_{\mathrm{sub} }(\mathbf{f}, \mathcal{C}_\mathrm{sub} \mid \mathcal{C}_\mathrm{sub} \neq \mathcal{C} ) - \\mathbb{E}\_\{ c, \\{ c\_k^- \\}^K\_\{k=1\} \} \ln (\mathrm{ \mathrm{Col}+1}) - 2d(\mathbf{f})  \right]$.
>
> That is mentioned in L284. If this explicit form of the upper bound of supervised loss makes Figure 1 clearer, we would like to add the equation and Arora et al's equation to Sec. 5.1 as follows:
> > Figure 1 shows the linear accuracy and _upper_ bounds of supervised loss $L_{\mathrm{sup}}$ by rearranging Equations (8) and (10): _[Eq. (8)'s upper bound]_ and _[the equation above]_, respectively.
>
> We would like to clarify the peak of our upper bound in Figure 1. It is rarely possible to draw all latent classes from smaller negative samples from the coupon collector’s problem. For example, $\upsilon_{K+1} \approx 4.5 \times 10^{-4}$ with $128$ from Eq. (9) for the CIFAR-100 experiment. This is why the upper bound has a peak there in Figure 1.

---

> > ### Comment · Reviewer_Rz1c · 2021-09-09
> > **One remaining concern**
> >
> > I am satisfied with the second two points, but still not sure about the first. From what I can tell, the idea used in both Arora et al. and this paper is that the latent classes exist on a per-instance level, rather than a coarser-grained level, allowing one to form the self-supervised problem without requiring annotations. This does not seem to agree with the example given by the authors in their response, but perhaps I am taking this example too literally?

---

> > > ### Author Response · Authors · 2021-09-09
> > > **Re: One remaining concern**
> > >
> > > Dear Reviewer Rz1c,
> > >
> > > We really appreciate that you are satisfied with our feedback to address most of the concerns and increase the rating. We would like to clarify the rest.
> > >
> > > We might misunderstand the initial question in Q1. Let me clarify it again. We think that there are two cases to say the supervised classes are mixtures/unions of latent classes as follows.
> > >
> > > - As we answered in the first response in https://openreview.net/forum?id=pZ5X_svdPQ&noteId=LWfITX-GIEq, latent classes are disjoint (breeds of dog and cat). In this case, the supervised class might be the union of them (dog and cat).
> > > - As clarified by Reviewer Rz1c, a latent class can overlap with others; a sample can be drawn from different latent classes. For example, disentangled properties of dog/cat such as breeds of dog and cat; their color, size, etc. The supervised classes might be “dog” and “cat”, or more fine-grained classes: “black dog” and “white cat” depending on the application.
> > >
> > > We think both cases are covered by our analysis because the feature representations are aggregated into the weights of the mean classifier by Jensen's inequality depending on the supervised class in our analysis. We would like to clarify the relationship between latent class and supervised class more in our manuscript. Precisely, we will address that the latent class can overlap in Sec. 3.1.1 and the supervised class is not only the subset of latent classes but also union or product of latent classes in Sec. 3.2.

---

> ### Author Response · Authors · 2021-09-08
> **Hope for rolling discussion**
>
> Dear Reviewer Rz1c,
>
> As in Reviewer Guidelines, this NeurIPS discussion is a rolling style to minimize the chance of misunderstandings on the submitted papers by reviewers. So we, the authors, would like to know whether or not we have addressed all concerns raised by the reviewer in our first feedback. However, we could not have seen additional feedback from the reviewer for this NeurIPS discussion period; we hope that the reviewer was happy with our author's feedback.
>
> But, in the first review, the reviewer wrote
>
> > I am open to revising this score upwards after reading feedback from the authors.
>
> Thus we really appreciate that the reviewer could revise the rating by reading our feedback. Otherwise, we would like to know the remaining concerns and clarify them.

---

### Author Response · Authors · 2021-09-09
**Analysis of Self-supervised learning really needs transfer learning setting?**

Dear reviewers QhKs and TgcC,

Both reviewers might still recognize not performing transfer learning settings as the main concern. We would like to argue that the analysis under the transfer learning setting is off-topic. Please recall that our main contribution is to reduce the mismatch between the empirical observation and theoretical analysis for a better understanding of self-supervised representation learning as in L37–41. Let us explain each of them a little bit more concretely.

### Empirical observations
As in the comprehensive experiments by Figure 9 in SimCLR and Figure 3 in MoCo, increasing the number of negative samples in self-supervised learning on the ImageNet-1K dataset improves the validation accuracy of the linear classifier with the learned feature encoder on the same ImageNet-1K dataset in practice. They do not discuss the tradeoff between the number of negative samples and performance under the transfer learning setting in our understanding.

###  Theoretical analysis
Arora et al. argue a large number of negative samples hurt the downstream performance, where the representation learning and supervised learning are done on the same dataset. Indeed Arora et al. wrote

> ​​a large number of negative samples could hurt. This problem, in fact, can arise even when the number of negative samples is much smaller than the number of classes.

Arora et al. demonstrate such empirical results in Figure D.1, where $K=10$’s performance is worse than $K=\\{2, 4\\}$’s one on the CIFAR-100 dataset. Clearly, we cannot explain the empirical observation reported by SimCLR and MoCo from Arora et al.’s theoretical analysis.

We believe that our analysis addresses the issue enough even though we do not analyze how the number of negative samples affects a transfer learning’s performance.
This is why the additional analysis under the transfer learning setting is off-topic in our work. Moreover, we would like to argue that if not performing theoretical analysis under transfer learning setting is the main reason for rejection, no existing theoretical work on self-supervised representation learning such as [2, 17, 18, 19, 20, 21] could be presented at machine learning conferences.

---

[17] Wang & Isola. Understanding Contrastive Representation Learning through Alignment and Uniformity on the Hypersphere. In _ICML_, 2020.

[18] Chuang et al. Debiased Contrastive Learning. In _NeurIPS_, 2020.

[19] Tosh et al. Contrastive Learning, Multi-view Redundancy, and Linear Models. In _ALT_, 2021.

[20] Wei et al. Theoretical Analysis of Self-Training with Deep Networks on Unlabeled Data. In _ICLR_, 2021

[21] Saunshi et al. A Mathematical Exploration of Why Language Models Help Solve Downstream Tasks. In _ICLR_, 2021.

---

### Decision · Program_Chairs · 2021-09-27

**Decision:**

Accept (Poster)

**Comment:**

This paper aims to understand a discrepancy between the theory literature (namely, Arora et al., 2019) and recent empirical results in self-supervised learning. Theory suggests that adding more negatives to contrastive learning should degrade downstream classification performance while empirical results have shown the contrary. To rectify this discrepancy, the authors expand prior analysis to introduce a new bound that is tighter in settings with large numbers of negatives.

Reviewers were conflicted about this paper. They all agreed that the problem is interesting and impactful, that the paper was clearly written, and that the contribution helps to resolve the conflict between theory and practice. There were also a number of technical concerns which were largely resolved through the discussion. However, several reviewers were concerned regarding the applicability of the theory to transfer settings in which the pre-training and downstream datasets differ since the current work only focuses on the case where these two datasets match. While I am sympathetic to this concern, I disagree that this addition is necessary for the paper to have impact or be a worthwhile contribution. Expanding the analysis to transfer settings is non-trivial and there is substantial value in addressing the case which is indeed de facto practice for many research papers on contrastive learning. I would encourage the authors, however, to consider the transfer setting for future work.